# Domain-Shift-Aware Conformal Prediction for Large Language Models

Zhexiao Lin [1]  Yuanyuan Li [2]  Neeraj Sarna [2]  Yuanyuan Gao [1]  Michael von Gablenz [2]

## Abstract

Large language models have achieved impressive performance across diverse tasks. However, their tendency to produce overconfident and factually incorrect outputs, known as hallucinations, poses risks in real-world applications. Conformal prediction provides finite-sample, distribution-free coverage guarantees, but standard conformal prediction breaks down under domain shift, often leading to under-coverage and unreliable prediction sets. We propose a new framework called Domain-Shift-Aware Conformal Prediction (DS-CP). Our framework adapts conformal prediction to large language models under domain shift, by systematically reweighting calibration samples based on their proximity to the test prompt, thereby preserving validity while enhancing adaptivity. Our theoretical analysis and experiments on the MMLU benchmark demonstrate that the proposed method delivers more reliable coverage than standard conformal prediction, especially under substantial distribution shifts, while maintaining efficiency. This provides a practical step toward trustworthy uncertainty quantification for large language models in real-world deployment.

## 1. Introduction

Large language models (LLMs) have rapidly advanced in recent years and are now widely deployed in real-world systems such as text summarization, information retrieval, and virtual assistants (Achiam et al., 2023; Zhang et al., 2020; Borgeaud et al., 2022; Wei et al., 2022). Despite their impressive performance, LLMs are prone to hallucination, generating outputs that are fluent and confident yet factually incorrect (Maynez et al., 2020; Ji et al., 2023; Huang et al., 2025). Such behavior poses risks in high-

stakes domains, ranging from healthcare and law to finance and scientific discovery, where inaccurate outputs can have significant consequences. To support safe and trustworthy deployment, it is therefore essential to ensure the reliability of LLM predictions (Bommasani, 2021; Weidinger et al., 2022). One promising direction is uncertainty quantification (UQ), which provides a principled way to measure and communicate the confidence of model outputs. UQ not only enables practitioners to better calibrate their trust in LLM responses but also serves as a foundation for downstream decision-making and risk mitigation in practical applications (Gawlikowski et al., 2023; Shorinwa et al., 2025).

One promising framework for UQ is conformal prediction (CP), which provides finite-sample and distribution-free coverage guarantees without requiring strong modeling assumptions (Vovk et al., 2005; Shafer & Vovk, 2008; Angelopoulos & Bates, 2023). Over the past decade, CP has been extensively developed as a general method for adding UQ to modern machine learning models, with applications spanning regression, classification, and structured prediction tasks (Lei et al., 2018; Angelopoulos et al., 2021; Romano et al., 2019). More recently, there has been growing interest in applying CP to LLMs, where it has been used to construct prediction sets for tasks such as text classification, multiple-choice question answering, and language generation (Kumar et al., 2023; Quach et al., 2024; Mohri & Hashimoto, 2024).

A key limitation of standard CP, however, is its reliance on the exchangeability assumption, which requires that calibration and test data are drawn from the same distribution (Tibshirani et al., 2019; Barber et al., 2023). This assumption is often violated in practice due to domain shift, a situation where the distribution of test data differs from that of the calibration data. Domain shift is pervasive in real-world applications because models are typically trained or calibrated on historical or benchmark datasets but deployed in dynamic environments where user behavior, input distributions, or data collection processes evolve over time (Moreno-Torres et al., 2012; Recht et al., 2019). If unaddressed, domain shift can lead to unreliable predictions and degraded UQ, undermining the reliability of machine learning systems in safety-critical settings such as healthcare, finance, and law (Ovadia et al., 2019; Gulrajani & Lopez-Paz, 2021). Empirical studies on LLMs further confirm this concern: when calibration and test distributions diverge, CP

[1]Department of Statistics, University of California, Berkeley, Berkeley, CA, USA [2]Munich Re, Munich, Germany. Correspondence to: Zhexiao Lin <zhexiaolin@berkeley.edu>, Yuanyuan Li <yli@munichre.com>.

*Proceedings of the 43rd International Conference on Machine Learning*, Seoul, South Korea. PMLR 306, 2026. Copyright 2026 by the author(s).

often under-covers, producing prediction sets that are too narrow and thus fail to achieve the intended coverage guarantees (Kumar et al., 2023). This makes the development of domain-shift-aware CP an essential step toward reliable and trustworthy deployment of LLMs in practice.

In this paper, we develop a new *domain-level adaptive* framework that extends CP to LLMs in the presence of domain shift. Our approach is motivated by the principles of CP under covariate shift (Tibshirani et al., 2019) and extensions beyond exchangeability (Barber et al., 2023), but adapts them to the unique challenges posed by LLMs. In particular, the high-dimensional and unstructured nature of prompts and responses makes it infeasible to directly estimate or model the underlying probability distribution. To address this, we leverage sentence embeddings to project prompts into a lower-dimensional semantic space that captures cross-domain similarity. Within this space, we apply a generalization of non-exchangeable CP, reweighting calibration samples according to their proximity to the prompt.

We evaluate our method on the MMLU benchmark (Hendrycks et al., 2021), a widely used benchmark for assessing LLM generalization across diverse domains. Our empirical results show that the proposed framework achieves more reliable coverage than standard CP, especially under substantial domain shifts where traditional CP is prone to severe under-coverage (Kumar et al., 2023; Ye et al., 2024). Importantly, our method strikes a balance between validity and adaptivity: it preserves rigorous statistical guarantees while tailoring prediction sets to the semantic structure of prompts.

**Contributions.** Our contributions are three-fold: (i) We introduce a novel CP framework specifically tailored for LLMs under domain shift by leveraging semantic embedding techniques to capture cross-domain similarity. (ii) We establish theoretical guarantees for our method, showing that it achieves valid coverage even when calibration and test distributions differ, thereby extending the reliability of CP beyond the standard exchangeability setting. (iii) Through experiments on the MMLU benchmark, we demonstrate that our approach consistently outperforms standard CP in terms of coverage under domain shift, while maintaining competitive efficiency and practical applicability in real-world deployment scenarios. Our contribution is a principled adaptation of conformal prediction under domain shift to LLM settings, where prompt distributions are high-dimensional and unstructured. The main novelty lies in combining semantic prompt embeddings with density-ratio-based weighting and a regularized data-dependent conformal construction that admits finite-sample coverage bounds and reduces to standard conformal prediction under exchangeability. Code is available at https://github.com/zhexiaolin/CP.

**Conflict of interest disclosure.** The authors declare no financial conflicts of interest related to this work.

## 2. Preliminaries

### 2.1. Setup

Suppose we have a pre-trained model $f : \mathcal{X} \to \mathcal{Y}$, which maps an input prompt to an output response. We observe prompt–ground truth pairs $(X_1, Y_1), \ldots, (X_n, Y_n)$ drawn exchangeably from an source domain. Given a new prompt $X_{n+1}$ sampled from a target domain, with corresponding (but unobserved) ground truth $Y_{n+1}$, our goal is to construct a prediction set $\hat{C}(X_{n+1}) \subset \mathcal{Y}$ using samples $\{(X_i, Y_i)\}_{i=1}^n$ such that, for a user-specified miscoverage level $\alpha \in (0, 1)$,

$$P(Y_{n+1} \in \hat{C}(X_{n+1})) \geq 1 - \alpha, \qquad (1)$$

which holds marginally for $(X_1, Y_1), \ldots, (X_n, Y_n)$ and $(X_{n+1}, Y_{n+1})$.

We assume access to labeled data $\{(X_i, Y_i)\}_{i=1}^n$ from the source domain, which can be used as calibration data. For the target domain, only prompts are available and ground truth labels are not observed. A main difficulty is that LLM prompts are in a vast, high-dimensional space $\mathcal{X}$ (token sequences with long-range dependencies), so even when one adopts a covariate shift perspective, estimating the density ratio from text is statistically and computationally infeasible. Moreover, the output space $\mathcal{Y}$ ranges from finite multiple-choice sets to open-ended natural language responses, which affects the choice of nonconformity score and how to measure "set size." Our goal is to design an uncertainty quantification procedure that maintains valid coverage guarantees for $Y_{n+1}$ despite the domain shift in prompts and the lack of labeled samples from the target domain.

### 2.2. Conformal Prediction

Consider a nonconformity score function $\mathcal{S} : \mathcal{X} \times \mathcal{Y} \to \mathbb{R}$, which measures the discrepancy between the ground truth $Y$ and the model prediction $f(X)$. Define the calibration scores $S_i = \mathcal{S}(X_i, Y_i)$ for $i = 1, \ldots, n$. We then introduce the standard CP, weighted CP, and nonexchangeable CP used in previous literature.

**Standard CP.** The standard CP constructs the empirical distribution of scores as

$$\hat{\mu}_{\mathrm{CP}} = \sum_{i=1}^n \frac{1}{n+1} \delta_{S_i} + \frac{1}{n+1} \delta_\infty,$$

where $\delta_s$ denotes the Dirac measure at $s \in \mathbb{R}$. For a new input covariate $x \in \mathcal{X}$, the prediction set $\hat{C}_{\mathrm{CP}}(x)$ is then

$$\hat{C}_{\mathrm{CP}}(x) = \{y \in \mathcal{Y} : \mathcal{S}(x, y) \leq \mathrm{Quantile}\,(1 - \alpha; \hat{\mu}_{\mathrm{CP}})\},$$

where $\mathrm{Quantile}(q; \cdot)$ denotes the $q$-th quantile of the input distribution.

The standard CP is a direct application of split CP when using a pretrained model, where there is no need to separate a training data for training prediction model. The validity of standard CP relies on the exchangeability assumption: $(X_1, Y_1), \ldots, (X_n, Y_n)$ and $(X_{n+1}, Y_{n+1})$ are exchangeable. This assumption fails under domain shift, since the test pair $(X_{n+1}, Y_{n+1})$ will not follow the same distribution as the calibration pairs $(X_1, Y_1), \ldots, (X_n, Y_n)$.

**Weighted CP.** When the input covariate space $\mathcal{X}$ is low dimensional, Tibshirani et al. (2019) proposed weighted CP. Let $\mathcal{P}_X$ and $\mathcal{P}'_X$ denote the marginal distributions of covariates of the calibration data and test data, respectively. Define the density ratio $r(x) = (d\mathcal{P}'_X / d\mathcal{P}_X)(x)$ for $x \in \mathcal{X}$. Let $r_i = r(X_i)$ for $i = 1, \ldots, n$ and $r_x = r(x)$. Weighted CP constructs the empirical distribution of scores as

$$\hat{\mu}_{\text{WCP}} = \sum_{i=1}^{n} \frac{r_i}{\sum_{j=1}^{n} r_j + r_x} \delta_{S_i} + \frac{r_x}{\sum_{j=1}^{n} r_j + r_x} \delta_{\infty}.$$

For a new input covariate $x \in \mathcal{X}$, the prediction set $\hat{C}_{\text{WCP}}(x)$ is then

$$\hat{C}_{\text{WCP}}(x) = \{y \in \mathcal{Y} : \mathcal{S}(x, y) \leq \text{Quantile}(1 - \alpha; \hat{\mu}_{\text{WCP}})\}.$$

This method guarantees valid coverage if the conditional distributions coincide across calibration and test data, i.e., $Y_{n+1} \mid X_{n+1}$ is the same as the conditional distribution $Y_i \mid X_i$ for $i = 1, \ldots, n$. When there is no covariate shift, the density ratio $r(x)$ is 1 for all $x \in \mathcal{X}$, which reduces to standard CP.

In practice, the density ratio $r(x)$ is unknown and must be estimated. When covariates from both calibration and test data are available, $r(x)$ can be estimated by training a classifier. Let $W = 0$ indicate an observation from the calibration data and $W = 1$ from the test data. Then we can write $r(x)$ as

$$r(x) = (d\mathcal{P}'_X / d\mathcal{P}_X)(x) = \frac{P(W = 1 \mid X = x)}{P(W = 0 \mid X = x)} \frac{P(W = 0)}{P(W = 1)}.$$

Thus, training a classifier to estimate $P(W = 1 \mid X = x)$ provides an estimate of $r(x)$.

While weighted CP performs well for low-dimensional covariates with covariate shift, its application to high-dimensional inputs (such as prompts in LLMs) is fundamentally limited. First, reliable density ratio estimation is statistically infeasible due to the curse of dimensionality, even with state-of-the-art classifiers (Wang et al., 2025). Second, the density ratio can be highly unbalanced, leading to the prediction set degenerating into the entire output space $\mathcal{Y}$, which is not informative although it is valid. Therefore, this motivates the need for dimension reduction and semantic-aware methods to adapt CP for LLMs with domain shift.

**Nonexchangeable CP.** For nonexchangeable data, Barber et al. (2023) proposed nonexchangeable CP. The nonex-

changeable CP constructs the empirical distribution of scores as

$$\hat{\mu}_{\text{NCP}} = \sum_{i=1}^{n} \frac{w_i}{\sum_{j=1}^{n} w_j + 1} \delta_{S_i} + \frac{1}{\sum_{j=1}^{n} w_j + 1} \delta_{\infty},$$

for some fixed weights $w_1, \ldots, w_n \in [0, 1]$. For a new input covariate $x \in \mathcal{X}$, the prediction set $\hat{C}_{\text{NCP}}(x)$ is then

$$\hat{C}_{\text{NCP}}(x) = \{y \in \mathcal{Y} : \mathcal{S}(x, y) \leq \text{Quantile}(1 - \alpha; \hat{\mu}_{\text{NCP}})\}.$$

The nonexchangeable CP is a general framework that can be applied to any nonexchangeable data, including both covariate shift and distribution shift. However, nonexchangeable CP cannot give $1 - \alpha$ coverage guarantee under either covariate shift or distribution shift (Theorem 2 in Barber et al. (2023)). The coverage discrepancy depends on the choice of weights $w_1, \ldots, w_n$, as well as the total variation distance between the calibration and test data, which cannot be estimated in practice as we do not have access to the ground truth of the test data. Choosing the weights is based on heuristic and remains an open question. There are some works applying this framework to LLMs with weights chosen by heuristic, e.g., Ulmer et al. (2024). This motivates the need for providing a systematic way to choose data-dependent weights and giving theoretical guarantees.

## 3. Method

In this section, we introduce our framework called domain-shift-aware conformal prediction (DS-CP).

**Embedding step.** To overcome the challenge of estimating density ratios in the high-dimensional prompt space $\mathcal{X}$, we leverage a pre-trained embedding model $h : \mathcal{X} \to \mathcal{Z}$, where $\mathcal{Z}$ is a finite-dimensional embedding space. The embedding model serves two purposes: (i) it reduces the dimensionality of the prompts, making statistical estimation more feasible; and (ii) it captures the semantic information of the prompts, so that prompts with similar meanings are mapped to nearby points in $\mathcal{Z}$. Therefore, the domain shift in prompts can be captured by the domain shift in the embedding space.

Let $\mathcal{P}_Z$ and $\mathcal{P}'_Z$ denote the source- and target-domain prompt distributions in embedding space. The density ratio in the embedding space is then $\tilde{r}(z) = (d\mathcal{P}'_Z / d\mathcal{P}_Z)(z)$. We carry out CP directly in the embedding space since the embedding preserves the semantic information of the prompts. Let $\tilde{r}_i = \tilde{r}(h(X_i))$ for $i = 1, \ldots, n$ and $\tilde{r}_x = \tilde{r}(h(x))$ for a new input prompt $x \in \mathcal{X}$. We first consider the weighted CP and thus the corresponding empirical distribution of scores is

$$\sum_{i=1}^{n} \frac{\tilde{r}_i}{\sum_{j=1}^{n} \tilde{r}_j + \tilde{r}_x} \delta_{S_i} + \frac{\tilde{r}_x}{\sum_{j=1}^{n} \tilde{r}_j + \tilde{r}_x} \delta_{\infty}.$$

This embedding-based approach provides a practical pathway to apply weighted CP under domain shift for LLMs, where directly estimating density ratios in the raw prompt space would be infeasible.

However, even after projecting prompts into a finite-dimensional embedding space, the dimension of $\mathcal{Z}$ remains relatively high in LLM applications, e.g., 384, 768, or 1024 for most BERT-based sentence-transformer models. As a result, when the shift between the source and target domains is large, the estimated density ratio can become highly unbalanced. In particular, we will have $\tilde{r}(h(x)) \gg \tilde{r}(h(X_i)), \quad i = 1, \ldots, n$, which implies that the empirical distribution places excessive mass on $\delta_\infty$. This effect is exacerbated when the calibration sample size $n$ is small, leading the resulting CP set to degenerate into the entire output space $\mathcal{Y}$. Such prediction sets are statistically valid but practically overly conservative, providing little useful information. We want to develop a method which is statistically valid but also practically efficient.

**Regularization step.** To mitigate this issue, we propose a regularization step: replace the test-point weight $\tilde{r}(h(x))$ by a regularized weight $\lambda = \lambda(X_1, \ldots, X_n) \geq 0$ which only depends on the calibration data $(X_1, \ldots, X_n)$. The proposed empirical distribution of scores becomes

$$\sum_{i=1}^{n} \frac{\tilde{r}(h(X_i))}{\sum_{j=1}^{n} \tilde{r}(h(X_j)) + \lambda} \delta_{S_i} + \frac{\lambda}{\sum_{j=1}^{n} \tilde{r}(h(X_j)) + \lambda} \delta_\infty.$$

This new empirical distribution has several key advantages: First, the distribution now depends solely on the calibration data $\{(X_i, Y_i)\}_{i=1}^{n}$, eliminating the instability caused by extreme test-point weights, and reducing the computational cost since we do not need to compute the empirical distribution for different test inputs. Second, when there is no domain shift ($\mathcal{P}'_Z = \mathcal{P}_Z$), the density ratios satisfy $\tilde{r}(h(X_i)) \equiv 1$ for all $i = 1, \ldots, n$, and the method reduces exactly to standard CP by taking $\lambda \equiv 1$. Third, by down-weighting the influence of extreme ratios from test point, the resulting prediction sets avoid trivial inflation to the entire output space, yielding more informative uncertainty quantification even under large domain shifts.

**Density ratio estimation.** In practice, the density ratio $\tilde{r}$ is unknown and must be estimated from data. We have the following relationship

$$\tilde{r}(z) = (\mathrm{d}\mathcal{P}'_Z/\mathrm{d}\mathcal{P}_Z)(z) = \frac{\mathrm{P}(W = 1 \mid Z = z)}{\mathrm{P}(W = 0 \mid Z = z)} \frac{\mathrm{P}(W = 0)}{\mathrm{P}(W = 1)}.$$

where $W = 0$ denotes the source domain and $W = 1$ denotes the target domain. This relationship suggests a natural estimation strategy: after embedding the prompts from both domains, we train a domain classifier to estimate $\mathrm{P}(W = 1 \mid Z = z)$. Any flexible machine learning classifier (e.g., logistic regression, gradient boosting, or neural networks) that works for high-dimensional data can be used for this classification task. By the fitted classifier and plugging into the previous equation, we obtain the estimated density ratio $\hat{r}(z)$. We can also directly estimate the density ratio by flexible density ratio estimation methods (Lin et al., 2023; Lin & Han, 2025; Cattaneo et al., 2025).

**Prediction set.** For each calibration point $X_i$, we define the estimated weight $\hat{w}_i = \hat{r}(h(X_i))$. With these estimated weights, the empirical distribution of scores becomes

$$\hat{\mu}_n = \sum_{i=1}^{n} \frac{\hat{w}_i}{\sum_{j=1}^{n} \hat{w}_j + \lambda} \delta_{S_i} + \frac{\lambda}{\sum_{j=1}^{n} \hat{w}_j + \lambda} \delta_\infty.$$

Finally, the prediction set for a new prompt $x \in \mathcal{X}$ is given by

$$\hat{C}_n(x) = \{y \in \mathcal{Y} : \mathcal{S}(x, y) \leq \mathrm{Quantile}\,(1 - \alpha; \hat{\mu}_n)\}.$$

**Algorithm.** The algorithm is summarized in Algorithm 1.

**Choosing the regularized weight $\lambda$.** In practice, the choice of the regularization parameter $\lambda$ controls the trade-off between conservativeness and adaptivity. A larger $\lambda$ down-weights the contribution of the calibration data more aggressively, leading to more conservative prediction sets, while a smaller $\lambda$ increases adaptivity but may risk under-coverage under severe domain shift. Setting $\lambda = \max_{i=1,\ldots,n} \hat{w}_i$ provides a principled balance between statistical validity and practical efficiency. First, this choice is directly supported by the theoretical guarantees in Section 4, which require $\lambda \geq \max_i \hat{w}_i$. Second, under the exchangeability condition, where $\hat{w}_i = 1$ for all $i = 1, \ldots, n$, this choice reduces DS-CP exactly to standard CP, thereby clarifying the connection between the two methods and enabling a fair comparison. More generally, the framework allows tuning $\lambda$ to reflect application-specific risk preferences. This regularization can be interpreted as a form of tempering or shrinkage, preventing the test-point weight from overwhelming the calibration distribution while preserving sensitivity to domain shift.

**Relationship to nonexchangeable CP.** The empirical score distribution used in DS-CP is closely related to the general class of nonexchangeable CP introduced by Barber et al. (2023). The key difference lies in how the weights are chosen. In nonexchangeable CP, weights are typically specified heuristically as priors to favor calibration points deemed "similar" to the test point, and the study of data-dependent weights was left as future work. In contrast, we propose a systematic data-dependent weighting scheme. The resulting weights depend on the whole calibration data through the density ratio estimation step and the definition of our empirical distribution of scores. For our method, calibration points with larger density ratios receive higher weights, reflecting that their embeddings are more representative of the target domain. Conversely, points with small density ratios contribute less, as they are less informative for predicting in the target domain. This construction naturally balances adaptivity and validity. When the calibration data has many high-weight points, which indicates the old and target domains are similar, the method behaves similarly to weighted CP and yields informative, data-adaptive prediction sets. On the other hand, when the weights for calibration data are

**Algorithm 1** Domain-Shift-Aware Conformal Prediction (DS-CP)

---

**Input:** Labeled source calibration data $\{(X_i, Y_i)\}_{i=1}^n$, unlabeled target-domain prompt batch $\mathcal{U} = \{\widetilde{X}_j\}_{j=1}^m$, embedding model $h : \mathcal{X} \to \mathcal{Z}$, nonconformity score $\mathcal{S}$, miscoverage level $\alpha \in (0, 1)$.

**Step 1: Embed source and target prompts.**
Compute $Z_i = h(X_i)$ for $i = 1, \ldots, n$ and $\widetilde{Z}_j = h(\widetilde{X}_j)$ for $j = 1, \ldots, m$.

**Step 2: Estimate density ratios in embedding space.**
Label source embeddings by $W = 0$ and target-batch embeddings by $W = 1$. Train a domain classifier to estimate $\hat{p}(z) \approx \mathrm{P}(W = 1 \mid Z = z)$, and set

$$\hat{r}(z) = \frac{\hat{p}(z)}{1 - \hat{p}(z)} \cdot \frac{\mathrm{P}(W = 0)}{\mathrm{P}(W = 1)}.$$

**Step 3: Form calibration weights and regularizer.**
Set $\hat{w}_i = \hat{r}(Z_i)$ for $i = 1, \ldots, n$, and choose $\lambda \geq 0$ (default: $\lambda = \max_{i=1,\ldots,n} \hat{w}_i$).

**Step 4: Compute calibration scores.**
Evaluate $S_i = \mathcal{S}(X_i, Y_i)$ for $i = 1, \ldots, n$.

**Step 5: Construct the target-domain-conditioned threshold.**
Form

$$\hat{\mu}_n = \sum_{i=1}^n \frac{\hat{w}_i}{\sum_{j=1}^n \hat{w}_j + \lambda} \delta_{S_i} + \frac{\lambda}{\sum_{j=1}^n \hat{w}_j + \lambda} \delta_\infty,$$

and compute $\hat{q}_{1-\alpha} = \mathrm{Quantile}(1 - \alpha; \hat{\mu}_n)$.

**Step 6: Predict on the target domain.**
For any prompt $x$ evaluated using the target batch $\mathcal{U}$, return

$$\hat{C}_n(x) = \{y \in \mathcal{Y} : \mathcal{S}(x, y) \leq \hat{q}_{1-\alpha}\}.$$

**Output:** The shared threshold $\hat{q}_{1-\alpha}$ and the target-domain-conditioned prediction rule $\hat{C}_n(\cdot)$.

---

uniformly low, indicating a large domain shift and limited relevance of the source domain, the procedure automatically down-weights unreliable calibration scores and becomes more conservative, reflecting the limited information available about the target domain.

## 4. Coverage Guarantees

We now analyze the coverage properties of DS-CP. Throughout this section, the estimated weights $\hat{w}_1, \ldots, \hat{w}_n$ and the regularizer $\lambda$ may be arbitrary measurable functions of the observed prompts (and any auxiliary randomness used to fit the domain classifier), but they do not depend on the unobserved target labels. For notational simplicity, define $\pi_i = \hat{w}_i/(\sum_{j=1}^n \hat{w}_j + \lambda)$ for $i = 1, \ldots, n$ and

$\pi_{n+1} = \lambda/(\sum_{j=1}^n \hat{w}_j + \lambda)$. Thus, $\pi_1, \ldots, \pi_n, \pi_{n+1}$ are the normalized masses placed on the calibration scores and on the point at infinity in the empirical score distribution. Let $Z_i = h(X_i)$ for $i = 1, \ldots, n + 1$, and write $\mathbf{S} = (S_1, \ldots, S_{n+1})$ for the vector of calibration scores together with the target score. For each $i \in \{1, \ldots, n\}$, let $\mathbf{S}^i = (S_1, \ldots, S_{i-1}, S_{n+1}, S_{i+1}, \ldots, S_n, S_i)$ denote the score vector obtained by swapping $S_i$ and $S_{n+1}$. We write $\mathrm{TV}(\cdot, \cdot)$ for total variation distance. The proofs are deferred to Appendix A.

**Theorem 4.1** (Lower bound). *Suppose $\lambda \geq \max_{i=1,\ldots,n} \hat{w}_i$ almost surely. Then*

$$\mathrm{P}\big(Y_{n+1} \in \hat{C}_n(X_{n+1})\big) \geq 1 - \alpha - \sum_{i=1}^n \mathrm{E}\big[\pi_i \, \mathrm{TV}(\mathbf{S}^i, \mathbf{S} \mid \mathcal{G})\big],$$

*where $\mathcal{G} = \sigma(\hat{w}_1, \ldots, \hat{w}_n, \lambda)$.*

Theorem 4.1 extends the nonexchangeable conformal analysis of Barber et al. (2023, Theorem 2) to our setting with data-dependent weights. It shows that DS-CP achieves the nominal level $1 - \alpha$ up to a coverage gap determined by two factors: the normalized weights $\pi_i$, which determine how strongly each calibration point influences the procedure, and the conditional total variation distances $\mathrm{TV}(\mathbf{S}^i, \mathbf{S} \mid \mathcal{G})$, which measure how different the score process is from the ideal exchangeable case after swapping the $i$th calibration score with the test score.

This lower bound is informative for two reasons. First, it makes clear that validity does not depend directly on the magnitude of the covariate shift in the raw prompt space, but rather on how that shift propagates to the nonconformity scores. Second, it shows why weighting helps: calibration points receiving small normalized mass $\pi_i$ contribute little to the coverage gap, while points that are more representative of the target domain can receive larger weight without sacrificing control unless their score distribution differs substantially from that of the test point.

Theorem 4.1 also recovers standard CP as a special case. If $\hat{w}_1 = \cdots = \hat{w}_n = 1$ and $\lambda = \max_{i=1,\ldots,n} \hat{w}_i = 1$, then DS-CP reduces exactly to standard CP, and $\mathcal{G}$ is trivial. If in addition $S_1, \ldots, S_{n+1}$ are exchangeable, then for every $i$ we have $\mathrm{TV}(\mathbf{S}^i, \mathbf{S} \mid \mathcal{G}) = \mathrm{TV}(\mathbf{S}^i, \mathbf{S}) = 0$. Hence the coverage gap vanishes and Theorem 4.1 reduces to $\mathrm{P}\big(Y_{n+1} \in \hat{C}_n(X_{n+1})\big) \geq 1 - \alpha$, which is exactly the usual finite-sample guarantee of standard CP. This confirms that DS-CP does not lose validity in the exchangeable setting; rather, it interpolates between standard CP and a domain-shift-aware weighted procedure.

Theorem 4.1 is stated in terms of a joint total variation distance on the full score vector. The next result shows that, under a conditional independence structure, this joint discrepancy can be reduced to a simpler comparison between the conditional laws of individual scores.

**Theorem 4.2.** *Assume that, almost surely, for each $i = 1, \ldots, n+1$, $S_i \perp (Z_1, \ldots, Z_{i-1}, Z_{i+1}, \ldots, Z_{n+1}) \mid Z_i$, and that $S_1, \ldots, S_{n+1}$ are conditionally independent given $Z_1, \ldots, Z_{n+1}$. Then*

$$\sum_{i=1}^{n} \mathrm{E}\big[\pi_i \mathrm{TV}(\mathbf{S}^i, \mathbf{S} \mid \mathcal{G})\big] \leq 2 \sum_{i=1}^{n} \mathrm{E}[\pi_i \mathrm{TV}(S_i \mid Z_i, S_{n+1} \mid Z_{n+1})]$$

Theorem 4.2 clarifies what aspect of domain shift matters for coverage. The coverage gap can be controlled by how different the one-dimensional conditional score distributions are at $Z_i$ and $Z_{n+1}$. This gives a clean interpretation: DS-CP is reliable whenever nearby or highly weighted prompts induce similar score distributions, even if the original prompts themselves come from substantially different domains.

The next theorem makes this idea quantitative through a smoothness condition in the embedding space.

**Theorem 4.3.** *Assume there exists a probability kernel $\{Q_z : z \in \mathcal{Z}\}$ such that $S_i \mid Z_i = z \sim Q_z$ almost surely for all $i = 1, \ldots, n+1$, and that for some metric $d(\cdot, \cdot)$ on $\mathcal{Z}$ and some constant $L > 0$, $\mathrm{TV}(Q_z, Q_{z'}) \leq Ld(z, z')$ for all $z, z' \in \mathcal{Z}$. Then*

$$\sum_{i=1}^{n} \mathrm{E}[\pi_i \mathrm{TV}(S_i \mid Z_i, S_{n+1} \mid Z_{n+1})] \leq L \sum_{i=1}^{n} \mathrm{E}[\pi_i d(Z_i, Z_{n+1})].$$

Combining Theorems 4.1, 4.2, and 4.3, we obtain the following qualitative picture. The coverage gap is small when the weighted calibration embeddings are close to the test embedding and, more fundamentally, when the score law varies smoothly across the embedding space. In other words, DS-CP does not require the raw prompt distributions to be close. What it requires is that the nonconformity score be stable with respect to the semantic representation.

This observation is particularly relevant for LLM applications. In high-dimensional text domains, prompts from different subjects or tasks may look very different at the input level, so classical covariate-shift arguments based on raw features can be pessimistic. However, if the embedding map places semantically similar prompts nearby, and if the score function behaves similarly on nearby embeddings, then the effective coverage gap can still be small. This is exactly the setting DS-CP is designed to exploit.

Conversely, the theorems also identify a failure mode. If the score distribution changes sharply across domains, then the total variation terms in Theorem 4.2 can be large, and the smoothness bound in Theorem 4.3 becomes loose. In that regime, the calibration data contains limited information about the target point, so no conformal method can be expected to remain both sharp and well-calibrated without becoming more conservative. Thus, the bounds correctly reflect the intrinsic difficulty of uncertainty quantification under severe domain shift.

We next complement the lower bound with an upper bound, which characterizes the extent to which DS-CP may be conservative.

**Theorem 4.4** (Upper bound). *Suppose $\lambda \geq \max_{i=1,\ldots,n} \hat{w}_i$ almost surely, and that the scores $S_1, \ldots, S_{n+1}$ are distinct almost surely. Then*

$$\mathrm{P}\big(Y_{n+1} \in \hat{C}_n(X_{n+1})\big)$$

$$< 1 - \alpha + \mathrm{E}[\pi_{n+1}] + \sum_{i=1}^{n} \mathrm{E}\big[\pi_i \mathrm{TV}(\mathbf{S}^i, \mathbf{S} \mid \mathcal{G})\big].$$

Theorem 4.4 shows that the same discrepancy term governing the lower bound also governs possible over-coverage. The only additional term is $\mathrm{E}[\pi_{n+1}]$, which is the expected mass assigned to the point at infinity. This term appears because the empirical score distribution used by DS-CP includes the regularization mass $\lambda$, and hence the resulting quantile can be inflated relative to the oracle exchangeable benchmark. Therefore, $\mathrm{E}[\pi_{n+1}]$ quantifies the built-in conservativeness induced by regularization. We now make $\pi_{n+1}$ more explicit.

**Proposition 4.5.** *Let $n_{\mathrm{eff}} = \left(\sum_{i=1}^{n} \hat{w}_i\right)^2 / \left(\sum_{i=1}^{n} \hat{w}_i^2\right)$. If $\lambda = \max_{i=1,\ldots,n} \hat{w}_i$, then*

$$\frac{1}{n_{\mathrm{eff}} + 1} \leq \frac{\lambda}{\sum_{j=1}^{n} \hat{w}_j + \lambda} \leq \frac{1}{\sqrt{n_{\mathrm{eff}}} + 1}.$$

Proposition 4.5 relates the conservativeness term $\pi_{n+1}$ to the effective sample size $n_{\mathrm{eff}}$. When the estimated weights are reasonably balanced, $n_{\mathrm{eff}}$ is large and the upper bound shows that the extra mass placed on $\infty$ is small, so the upper bound in Theorem 4.4 stays close to the target level. Conversely, when only a few calibration points receive most of the weight, $n_{\mathrm{eff}}$ becomes small. In that regime the proposition shows that a non-negligible regularization mass is unavoidable, reflecting the fact that only a limited amount of target-relevant calibration information is available from the source domain. This behavior is desirable: the method automatically becomes more conservative precisely when the effective calibration sample size collapses.

## 5. Experiments

In this section, we evaluate the performance of our framework DS-CP on datasets exhibiting domain shift.

**Dataset.** We conduct experiments on the MMLU benchmark (Hendrycks et al., 2021), which has become a widely used benchmark for evaluating UQ for LLMs (Kumar et al., 2023; Ye et al., 2024; Kiyani et al., 2024; Su et al., 2024; Vishwakarma et al., 2025). Following the setup in Ye et al. (2024), we focus on the multiple-choice question answering task. The MMLU dataset spans 17 subjects grouped into four broad categories: humanities, social sciences, STEM, and other domains (business, health, and miscellaneous).

Each category contains 2,500 instances. For each question, there are six possible answer choices (A–F), exactly one of which is correct. This structure makes MMLU particularly well-suited for assessing calibrated prediction sets, since coverage and set size can be measured in a straightforward and interpretable way.

**Evaluation.** To study performance under domain shift, we treat different subjects as different domains. Specifically, we calibrate on one subject and test on another. With 17 subjects in total, this leads to $17 \times 16 = 272$ ordered subject pairs, since calibration and test domains are not interchangeable. We apply our method to all 272 pairs, which allows us to systematically evaluate robustness across a broad spectrum of domain shifts. Each sample corresponds to a subject pair. For each pretrained model, we report (i) the empirical coverage, defined as the proportion of test instances where the prediction set contains the correct answer, and (ii) the average set size, defined as the mean cardinality of the prediction sets across the test data. This dual evaluation captures the trade-off between validity and efficiency.

To properly assess marginal coverage, one would ideally resample both calibration and test data. Accordingly, for each subject pair, we conducted a bootstrap procedure by resampling calibration and test instances with replacement and recomputing empirical coverage across replicates. The resulting mean and median coverage were nearly identical to those obtained using the full calibration and test sets. This outcome is expected: conditional on the observed data, bootstrap averages converge to the full-sample statistic as the number of replicates increases. For this reason, and for computational clarity, we report results based on the full calibration and test data, which effectively capture the same behavior as the resampled evaluation.

**Parameters.** Following the previous discussion on the regularization parameter $\lambda$, we set the regularization parameter to $\lambda = \max_{i=1,\ldots,n} \hat{w}_i$ throughout our experiments. An ablation study on the choice of $\lambda$ is provided in Appendix B. Specifically, we vary $\lambda = \gamma \max_{i=1,\ldots,n} \hat{w}_i$ with $\gamma \in \{0, 0.25, 0.5, 1, 2, 5\}$, and show that $\lambda = \max_{i=1,\ldots,n} \hat{w}_i$ achieves a balance between validity and efficiency. We fix the miscoverage level at $\alpha = 0.10$, corresponding to a target marginal coverage of 90%.

**Models.** We evaluate our framework using a diverse set of 16 open-source LLMs drawn from 9 representative model families, following Ye et al. (2024). These include the Llama-2 series, Mistral-7B, Falcon series, MPT-7B, Gemma-7B, Qwen series, Yi series, DeepSeek series, and InternLM-7B. This collection spans a range of parameter scales, architectures, and training paradigms, ensuring that our evaluation is not tied to a single model family. For the embedding model, we adopt the `all-MiniLM-L6-v2` SentenceTransformer, which maps input questions into a 384-dimensional semantic space. To estimate density ratios, we train an XGBoost classifier on the embedded data, distinguishing calibration from test prompts. An ablation study on the choices of embedding model and classifier is provided in Appendix B.

**Score function.** For each LLM, we extract logits over the six answer choices $A, \ldots, F$ using the same prompting strategy following Ye et al. (2024). The logits are transformed into probabilities via the softmax function. We adopt the Least Ambiguous Classifier (LAC) nonconformity score, defined as $\mathcal{S}(X, Y) = 1 - f(X)_Y$, where $f(X)_Y$ denotes the softmax probability assigned to the label $Y$. Intuitively, this score reflects the model's uncertainty about the answer: smaller values indicate higher confidence. In Appendix B, we also evaluate with the Adaptive Prediction Sets (APS) score given by $\mathcal{S}(X, Y) = \sum_{Y' \in \mathcal{Y}} f(X)_{Y'} \mathbb{1}(f(X)_{Y'} > f(X)_Y)$, which ranks the label $Y$ against competing alternatives and accumulates probability mass over more likely labels.

**Empirical coverage.** Figure 1 compares the empirical coverage of standard CP (we denote it as CP in the figure) and our DS-CP across all evaluated models. For standard CP, we observe that coverage often falls short of the target 90% level. In fact, for every model there exist subject pairs where the empirical coverage drops well below 90%, and for several models, the median coverage itself lies below 90%. This implies that in more than half of the subject pairs, standard CP under-covers, highlighting its inability to maintain valid guarantees under domain shift. By contrast, DS-CP consistently improves coverage: across all models, its median coverage is higher than that of standard CP, and the lower tail of the distribution (below 90%) is substantially reduced. These results demonstrate that DS-CP is not only more robust to domain shift but also provides more reliable coverage in practice, mitigating the systematic under-coverage that plagues standard CP.

**Set size.** Figure 2 reports the average prediction set size for standard CP and DS-CP across all evaluated models. As expected, DS-CP generally produces larger sets than standard CP. This increase is natural: to address the under-coverage of standard CP under domain shift, DS-CP must enlarge its prediction sets to achieve the desired 90% coverage level. The magnitude of the set sizes varies across models, reflecting differences in their model predictive accuracy. More accurate predictive models tend to produce smaller sets overall. Importantly, while DS-CP does increase set size relative to standard CP, the increase is modest across all models. This demonstrates that DS-CP is able to remain efficient and restore valid coverage under domain shift, showing a practical balance between robustness and efficiency of the prediction sets.

**Does DS-CP solve the domain shift problem?** A natural

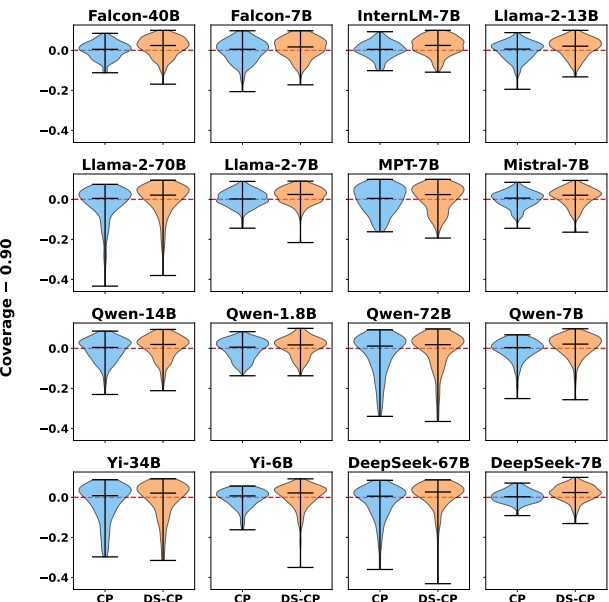

*Figure 1.* Empirical coverage of CP vs. DS-CP across models. The center bar is the median.

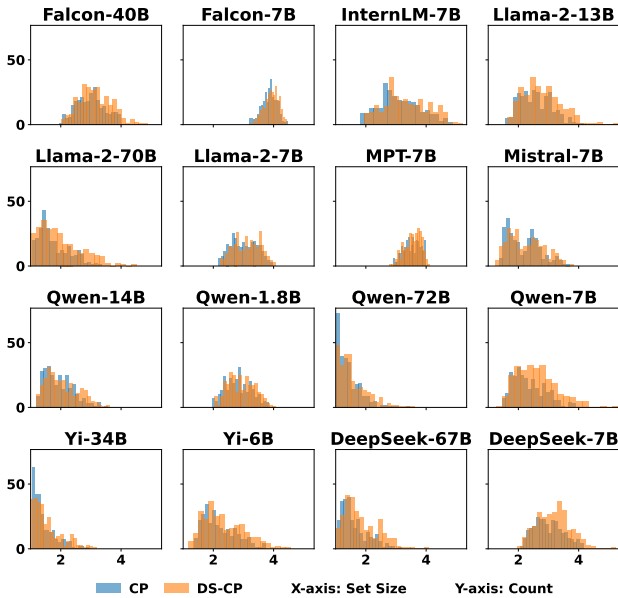

*Figure 2.* Average prediction set size of CP vs. DS-CP across models.

question is whether DS-CP genuinely addresses domain shift, or whether it merely enlarges prediction sets uniformly regardless of the input of the test point. To investigate this, Figure 3 presents the scatter plot of the empirical coverage of DS-CP versus standard CP across all subject pairs and models. Each point corresponds to one calibration-test pair: orange points mark cases where standard CP under-covers (coverage below the 90% target), while blue points denote cases where standard CP already achieves valid coverage. The diagonal $y = x$ line represents parity between DS-CP and CP.

The results show a clear pattern. For nearly all models, the majority of orange points lie above the diagonal, indicating that DS-CP systematically improves coverage precisely in the settings where standard CP fails. Moreover, the farther a point lies to the left (severe under-coverage by CP), the larger its vertical distance above the diagonal, showing that DS-CP provides the greatest improvements when the coverage gap is most severe. In contrast, the blue points on the right cluster close to the diagonal, suggesting that when standard CP already achieves valid coverage, DS-CP makes only minimal adjustments.

Together, these findings highlight that DS-CP is adaptive to the presence of domain shift: it selectively corrects under-coverage in difficult cases without excessively inflating coverage where standard CP is already valid. This adaptivity provides strong evidence that DS-CP goes beyond a naive set enlargement and genuinely mitigates the effects of domain shift.

## 6. Related Work

**Conformal prediction for LLMs.** Several recent works focus on adapting CP to practical constraints in LLM settings. CP has been applied to multiple-choice question answering (Kumar et al., 2023), to uncertainty-aware decision-making in LLM-based planners (Ren et al., 2023), and to benchmarking LLMs through uncertainty quantification (Ye et al., 2024). Su et al. (2024) develops CP methods that operate without access to model logits, making them suitable for closed-source LLM APIs. Vishwakarma et al. (2025) studies efficiency-oriented CP procedures that reduce prediction set size through structured pruning. Other works explore extensions of CP for language generation or factuality control (Quach et al., 2024; Mohri & Hashimoto, 2024; Wang et al., 2024). While these approaches demonstrate the flexibility of CP in LLM applications, they all fundamentally rely on the exchangeability assumption between calibration and test data and primarily aim to improve usability, efficiency, or task-specific reliability. As a result, their coverage guarantees do not adapt when domain shift is present, leaving the robustness of CP under domain shift unaddressed, which is the central focus of our work.

**Conformal prediction beyond exchangeability in LLMs.** A few recent works have begun to explore conformal prediction for LLMs beyond the standard exchangeable setting. Ulmer et al. (2024) apply nonexchangeable CP to LLMs using heuristic similarity-based weights, but their method lacks theoretical coverage guarantees and relies on ad hoc choices of similarity metrics. Cherian et al. (2024) considers CP under covariate shift, but requires explicitly specifying

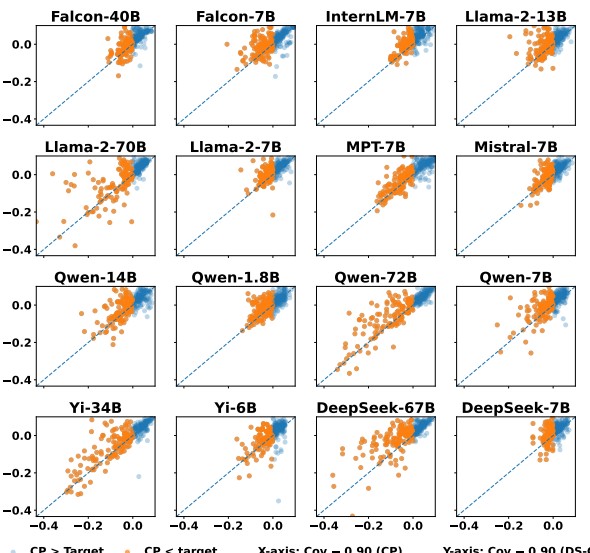

*Figure 3.* Paired coverage comparison of CP and DS-CP across models.

the shift mechanism, which is infeasible in practice for high-dimensional and unstructured LLM prompts. In contrast, our method is explicitly designed to handle domain shift in LLMs by introducing a principled, density-ratio-based weighting scheme defined in a semantic embedding space, yielding theoretical coverage guarantees while naturally recovering standard CP under exchangeability.

## 7. Discussion

DS-CP should be viewed as a target-domain-conditioned recalibration procedure for LLMs under domain shift. It combines semantic embeddings, density-ratio estimation, and regularization to produce a shared conformal threshold for a target-domain batch. Our theoretical results provide finite-sample lower and upper coverage bounds around the nominal level, and our experiments on a processed MMLU-derived benchmark show that this domain-level adaptation materially improves coverage relative to standard CP in the regimes where exchangeability fails most visibly.

At the same time, several limitations are important. First, DS-CP depends on the quality of the embedding model and of the density-ratio estimator; misspecification at either stage can reduce adaptivity or efficiency. Second, the method requires access to unlabeled target-domain prompts. This is a realistic assumption in many domain-adaptation settings, but it is stronger than the assumptions of purely online single-prompt prediction. Third, under severe shift, the method may return larger sets, and these larger sets should not be misread as evidence that the underlying model has become intrinsically reliable. Fourth, our experiments focus on a transductive multiple-choice benchmark, where

set size and coverage are easy to measure. Extending the framework to open-ended generation will require new score functions and evaluation protocols. Finally, our empirical study centers on standard CP as the main baseline; broader comparisons against additional domain-shift-aware conformal baselines remain an important direction for future work. In deployment, most of the additional computation is front-loaded into an offline adaptation stage: the embedding model and domain classifier are applied once to the source calibration prompts and the unlabeled target-domain batch, after which the conformal threshold can be computed and reused across the batch. Relative to LLM inference itself, this overhead is modest in our implementation, but the quality of the resulting threshold still depends on having enough unlabeled target-domain prompts to estimate the source–target shift reliably.

Several extensions are especially promising. One is an online or streaming variant that periodically refreshes the density-ratio model as new unlabeled prompts arrive; this would connect DS-CP to adaptive conformal methods, but would require new theory because the weights would evolve over time. Another is evaluation under temporal and stylistic shift, such as knowledge staleness or paraphrase variation. A third is combining DS-CP with complementary calibration methods such as temperature scaling. Since DS-CP only requires a nonconformity score as input, such combinations are conceptually natural.

## Acknowledgments

This work was primarily conducted during Z.L.'s internship at Munich Re. The authors thank the area chairs and four anonymous reviewers for their helpful discussions.

## Impact Statement

This paper presents work whose goal is to advance the field of Machine Learning. There are many potential societal consequences of our work, none which we feel must be specifically highlighted here.

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

## A. Proofs

For a probability measure $\nu$ on $\mathbb{R}$, define its $(1-\alpha)$-quantile by

$$q_{1-\alpha}(\nu) = \inf\{t \in \mathbb{R} : \nu((-\infty, t]) \geq 1 - \alpha\}.$$

### A.1. Proof of Theorem 4.1

*Proof.* Let

$$A = \{Y_{n+1} \notin \hat{C}_n(X_{n+1})\} = \left\{ S_{n+1} > q_{1-\alpha}\left( \sum_{j=1}^{n} \pi_j \delta_{S_j} + \pi_{n+1} \delta_\infty \right) \right\}.$$

Fix a realization of $\mathcal{G} = \sigma(\hat{w}_1, \ldots, \hat{w}_n, \lambda)$ for the moment. Conditional on $\mathcal{G}$, the numbers $\pi_1, \ldots, \pi_{n+1}$ are deterministic and satisfy $\pi_{n+1} \geq \max_{i \leq n} \pi_i$ because $\lambda \geq \max_{i \leq n} \hat{w}_i$.

For $u = (u_1, \ldots, u_{n+1}) \in \mathbb{R}^{n+1}$, define

$$\mu_{\text{fin}}(u) = \sum_{j=1}^{n} \pi_j \delta_{u_j} + \pi_{n+1} \delta_{u_{n+1}}, \qquad \mathcal{F}(u) = \{i \in \{1, \ldots, n+1\} : u_i > q_{1-\alpha}(\mu_{\text{fin}}(u))\}.$$

By the definition of the quantile, the mass strictly above $q_{1-\alpha}(\mu_{\text{fin}}(u))$ is at most $\alpha$, hence

$$\sum_{i \in \mathcal{F}(u)} \pi_i \leq \alpha \qquad \text{for every } u \in \mathbb{R}^{n+1}.$$

For each $i \in \{1, \ldots, n+1\}$, let

$$\mu_i = \mu_{\text{fin}}(\mathbf{S}^i) = \sum_{j=1}^{n} \pi_j \delta_{S_j^i} + \pi_{n+1} \delta_{S_{n+1}^i}.$$

We claim that, for every $i \leq n$,

$$q_{1-\alpha}(\mu_i) \leq q_{1-\alpha}\left( \sum_{j=1}^{n} \pi_j \delta_{S_j} + \pi_{n+1} \delta_\infty \right).$$

Indeed, for any finite $t$,

$$\mu_i((-\infty, t]) - \left( \sum_{j=1}^{n} \pi_j \delta_{S_j} + \pi_{n+1} \delta_\infty \right)((-\infty, t]) = \pi_i \left( \mathbb{1}\{S_{n+1} \leq t\} - \mathbb{1}\{S_i \leq t\} \right) + \pi_{n+1} \mathbb{1}\{S_i \leq t\}.$$

If $S_i > t$, then the right-hand side equals $\pi_i \mathbb{1}\{S_{n+1} \leq t\} \geq 0$. If $S_i \leq t$, then the right-hand side equals $\pi_i \mathbb{1}\{S_{n+1} \leq t\} - \pi_i + \pi_{n+1} \geq 0$ because $\pi_{n+1} \geq \pi_i$. Thus the cumulative distribution function of $\mu_i$ dominates that of $\sum_{j=1}^{n} \pi_j \delta_{S_j} + \pi_{n+1} \delta_\infty$ at every finite $t$, which proves the claim.

On the event $A$, we therefore have for every $i \leq n$,

$$S_{n+1} > q_{1-\alpha}\left( \sum_{j=1}^{n} \pi_j \delta_{S_j} + \pi_{n+1} \delta_\infty \right) \geq q_{1-\alpha}(\mu_i).$$

Since $S_{n+1} = S_i^i$, this means $i \in \mathcal{F}(\mathbf{S}^i)$ for every $i \leq n$. For $i = n+1$, we have $\mathbf{S}^{n+1} = \mathbf{S}$ and therefore $A$ also implies $n+1 \in \mathcal{F}(\mathbf{S})$.

Now introduce an auxiliary index $K$ such that, conditional on $\mathcal{G}$,

$$K \sim \sum_{i=1}^{n+1} \pi_i \delta_i,$$

and $K$ is independent of $\mathbf{S}$ given $\mathcal{G}$. The previous paragraph shows that

$$A \subseteq \{K \in \mathcal{F}(\mathbf{S}^K)\}.$$

Hence,

$$\mathrm{P}(A|\mathcal{G}) \leq \mathrm{P}(K \in \mathcal{F}(\mathbf{S}^K)|\mathcal{G}) = \sum_{i=1}^{n+1} \pi_i \, \mathrm{P}\big(i \in \mathcal{F}(\mathbf{S}^i)|\mathcal{G}\big).$$

For each $i$, let $B_i = \{u \in \mathbb{R}^{n+1} : i \in \mathcal{F}(u)\}$. Then $B_i$ is measurable with respect to the weights fixed by $\mathcal{G}$, and

$$\mathrm{P}\big(i \in \mathcal{F}(\mathbf{S}^i)|\mathcal{G}\big) = \mathrm{P}(\mathbf{S}^i \in B_i|\mathcal{G}) \leq \mathrm{P}(\mathbf{S} \in B_i|\mathcal{G}) + \mathrm{TV}(\mathbf{S}^i, \mathbf{S}|\mathcal{G}).$$

Therefore,

$$\begin{aligned}
\mathrm{P}(A|\mathcal{G}) &\leq \sum_{i=1}^{n+1} \pi_i \mathrm{P}(\mathbf{S} \in B_i|\mathcal{G}) + \sum_{i=1}^{n+1} \pi_i \, \mathrm{TV}(\mathbf{S}^i, \mathbf{S}|\mathcal{G}) \\
&= \mathrm{E}\left[\sum_{i=1}^{n+1} \pi_i \mathbb{1}\{i \in \mathcal{F}(\mathbf{S})\}\Big|\mathcal{G}\right] + \sum_{i=1}^{n+1} \pi_i \, \mathrm{TV}(\mathbf{S}^i, \mathbf{S}|\mathcal{G}) \\
&\leq \alpha + \sum_{i=1}^{n+1} \pi_i \, \mathrm{TV}(\mathbf{S}^i, \mathbf{S}|\mathcal{G}).
\end{aligned}$$

Since $\mathbf{S}^{n+1} = \mathbf{S}$, the last term vanishes for $i = n + 1$, and so

$$\mathrm{P}(A|\mathcal{G}) \leq \alpha + \sum_{i=1}^{n} \pi_i \, \mathrm{TV}(\mathbf{S}^i, \mathbf{S}|\mathcal{G}).$$

Taking expectations and using $\mathrm{P}(Y_{n+1} \in \hat{C}_n(X_{n+1})) = 1 - \mathrm{P}(A)$ gives the result. $\qquad\square$

### A.2. Proof of Theorem 4.2

*Proof.* Let

$$\mathcal{H} = \sigma(Z_1, \ldots, Z_{n+1}, \mathcal{G}).$$

Because the weights are covariate-only quantities, conditioning on $\mathcal{H}$ does not alter the conditional score laws once the corresponding embeddings are fixed. Define the random probability measures

$$P_i = \mathcal{L}(S_i|\mathcal{H}) = \mathcal{L}(S_i|Z_i), \qquad i = 1, \ldots, n+1.$$

Under the stated conditional independence assumptions,

$$\mathcal{L}(\mathbf{S}|\mathcal{H}) = P_1 \otimes \cdots \otimes P_n \otimes P_{n+1},$$

and, for each $i \leq n$,

$$\mathcal{L}(\mathbf{S}^i|\mathcal{H}) = P_1 \otimes \cdots \otimes P_{i-1} \otimes P_{n+1} \otimes P_{i+1} \otimes \cdots \otimes P_n \otimes P_i.$$

We first compare conditioning on $\mathcal{G}$ and on $\mathcal{H}$. For any Borel set $A \subseteq \mathbb{R}^{n+1}$,

$$\Big|\mathrm{P}(\mathbf{S}^i \in A|\mathcal{G}) - \mathrm{P}(\mathbf{S} \in A|\mathcal{G})\Big| = \Big|\mathrm{E}\Big[\mathrm{P}(\mathbf{S}^i \in A|\mathcal{H}) - \mathrm{P}(\mathbf{S} \in A|\mathcal{H})\Big|\mathcal{G}\Big]\Big| \leq \mathrm{E}\big[\mathrm{TV}(\mathbf{S}^i, \mathbf{S}|\mathcal{H})|\mathcal{G}\big].$$

Taking the supremum over $A$ yields

$$\mathrm{TV}(\mathbf{S}^i, \mathbf{S}|\mathcal{G}) \leq \mathrm{E}\big[\mathrm{TV}(\mathbf{S}^i, \mathbf{S}|\mathcal{H})|\mathcal{G}\big].$$

Conditional on $\mathcal{H}$, the two product measures differ only in coordinates $i$ and $n + 1$, and tensorization with a common factor preserves total variation. Hence

$$\mathrm{TV}(\mathbf{S}^i, \mathbf{S}|\mathcal{H}) = \mathrm{TV}(P_i \otimes P_{n+1}, P_{n+1} \otimes P_i).$$

Using the triangle inequality and invariance of total variation under tensoring by a fixed measure,

$$\mathrm{TV}(P_i \otimes P_{n+1}, P_{n+1} \otimes P_i) \leq \mathrm{TV}(P_i \otimes P_{n+1}, P_i \otimes P_i) + \mathrm{TV}(P_i \otimes P_i, P_{n+1} \otimes P_i)$$
$$= \mathrm{TV}(P_{n+1}, P_i) + \mathrm{TV}(P_i, P_{n+1}) = 2\,\mathrm{TV}(P_i, P_{n+1}).$$

Combining this with $\mathrm{TV}(\mathbf{S}^i, \mathbf{S}\,|\,\mathcal{G}) \leq \mathrm{E}\big[\mathrm{TV}(\mathbf{S}^i, \mathbf{S}\,|\,\mathcal{H})\,|\,\mathcal{G}\big]$, multiplying by $\pi_i$, and taking expectations gives

$$\mathrm{E}\big[\pi_i\,\mathrm{TV}(\mathbf{S}^i, \mathbf{S}\,|\,\mathcal{G})\big] \leq 2\mathrm{E}\big[\pi_i\,\mathrm{TV}(P_i, P_{n+1})\big].$$

Summing over $i = 1, \ldots, n$ proves the claim. $\qquad\square$

### A.3. Proof of Theorem 4.3

*Proof.* Under the assumptions of the theorem, for each $i \leq n$,

$$\mathrm{TV}(S_i\,|\,Z_i, S_{n+1}\,|\,Z_{n+1}) = \mathrm{TV}(Q_{Z_i}, Q_{Z_{n+1}}) \leq L\,d(Z_i, Z_{n+1}) \qquad \text{almost surely.}$$

Multiplying by $\pi_i$, taking expectations, and summing over $i = 1, \ldots, n$ yields

$$\sum_{i=1}^{n} \mathrm{E}\big[\pi_i\,\mathrm{TV}(S_i\,|\,Z_i, S_{n+1}\,|\,Z_{n+1})\big] \leq L \sum_{i=1}^{n} \mathrm{E}\big[\pi_i d(Z_i, Z_{n+1})\big].$$

$\qquad\square$

### A.4. Proof of Theorem 4.4

*Proof.* The proof of Theorem 4.4 is similar to the proof of Theorem 4.1, which generalizes Theorem 3 in (Barber et al., 2023). $\qquad\square$

### A.5. Proof of Proposition 4.5

*Proof.* Let

$$S = \sum_{i=1}^{n} \hat{w}_i, \qquad Q = \sum_{i=1}^{n} \hat{w}_i^2, \qquad M = \lambda = \max_{i=1,\ldots,n} \hat{w}_i.$$

Then $n_{\mathrm{eff}} = S^2/Q$ and

$$\pi_{n+1} = \frac{M}{S+M} = \frac{M/S}{1+M/S}.$$

Because each $\hat{w}_i \leq M$, we have $Q \leq MS$. Hence

$$n_{\mathrm{eff}} = \frac{S^2}{Q} \geq \frac{S^2}{MS} = \frac{S}{M},$$

which implies $M/S \geq 1/n_{\mathrm{eff}}$. Therefore,

$$\pi_{n+1} = \frac{M/S}{1+M/S} \geq \frac{1/n_{\mathrm{eff}}}{1+1/n_{\mathrm{eff}}} = \frac{1}{n_{\mathrm{eff}}+1}.$$

On the other hand, $Q \geq M^2$, so

$$n_{\mathrm{eff}} = \frac{S^2}{Q} \leq \frac{S^2}{M^2} = \left(\frac{S}{M}\right)^2.$$

Thus $M/S \leq 1/\sqrt{n_{\mathrm{eff}}}$, and consequently

$$\pi_{n+1} = \frac{M/S}{1+M/S} \leq \frac{1/\sqrt{n_{\mathrm{eff}}}}{1+1/\sqrt{n_{\mathrm{eff}}}} = \frac{1}{\sqrt{n_{\mathrm{eff}}}+1}.$$

This proves the proposition. $\qquad\square$

# B. Additional Experiments

Figure 4 presents the number of instances in each of the 17 subjects in the MMLU benchmark. Although the subjects are approximately balanced overall, moderate variation in sample size exists across domains, which further motivates the need for domain-shift-aware calibration methods that remain stable under heterogeneous data availability.

**Ablation study on the regularization parameter $\lambda$.** We conduct an ablation study to examine the effect of the regularization parameter $\lambda$. Specifically, we vary $\lambda = \gamma \max_{i=1,...,n} \hat{w}_i$ with $\gamma \in \{0, 0.25, 0.5, 1, 2, 5\}$ and report empirical coverage and average prediction set size aggregated across all subject pairs for all models. Figure 5 shows the empirical coverage, and Figure 6 shows the corresponding average set size. In both figures, the three curves for each method correspond to the minimum, median, and maximum values computed over all subject pairs.

We observe a clear monotonic trend: increasing $\gamma$ improves, or at least does not decrease, empirical coverage, but at the cost of larger prediction sets. When $\gamma < 1$, the method becomes less conservative and may under-cover for subject pairs exhibiting substantial domain shift. This regime can also violate the sufficient condition $\lambda \geq \max_i \hat{w}_i$ required by our theoretical analysis. In contrast, when $\gamma > 1$, the procedure becomes overly conservative: coverage gains saturate, while prediction sets inflate rapidly, approaching the trivial behavior induced by placing excessive mass on the point at infinity. Overall, $\gamma = 1$ (i.e., $\lambda = \max_i \hat{w}_i$) consistently achieves the best balance between validity and efficiency across models, aligning with our theoretical guidance and supporting its use as a robust default in practice.

**Ablation study on embedding models and classifiers.** We next examine the sensitivity of DS-CP to the choice of embedding model and classifier. To isolate the effect of the embedding model, we compare MiniLM, MPNet, and E5 while holding the classifier fixed. Conversely, to isolate the effect of the classifier, we compare XGBoost, a multilayer perceptron (MLP), and logistic regression (LR) while holding the embedding model fixed. For each setting, we report empirical coverage and average prediction set size, aggregated across all subject pairs. For the embedding-model ablation, Figure 7 shows the distribution of empirical coverage relative to the target level of 0.90, and Figure 8 shows the corresponding distribution of prediction set sizes. For the classifier ablation, Figure 9 and Figure 10 present the analogous coverage and set-size distributions, respectively.

Across the evaluated LLMs, DS-CP exhibits qualitatively similar behavior under all three embedding models and all three classifiers. The empirical coverage distributions are generally centered above zero relative to the target level, indicating that DS-CP maintains the desired coverage for most subject pairs. Moreover, the lower tails are comparable across both embedding models and classifiers, suggesting that no particular model choice substantially weakens coverage. The prediction set-size histograms also overlap considerably, indicating that the observed coverage behavior is not driven by any single embedding model or classifier producing systematically larger prediction sets. Taken together, these results suggest that DS-CP is reasonably robust to standard off-the-shelf choices of sentence embedding models and classifiers.

**APS score.** We further evaluate our framework using the APS score, following the same experimental setup as in the main text. Figure 11 compares the empirical coverage of standard CP and DS-CP across all evaluated models, while Figure 12 compares the corresponding average prediction set sizes. Figure 13 presents a scatter plot comparing DS-CP and standard CP coverage across all subject pairs and models.

Consistent with the results obtained using the LAC score, APS-based experiments show that DS-CP effectively corrects under-coverage caused by domain shift. In particular, DS-CP substantially improves coverage for subject pairs where standard CP fails, while maintaining prediction set sizes that remain practical. These results demonstrate that the benefits of DS-CP are not specific to a particular nonconformity score, but extend across commonly used CP constructions for LLM tasks.

# C. Experiment Details

We provide additional details on the implementation of the experiments.

**Dataset availability.** All MMLU data and logits for all evaluated models are reused from `https://github.com/smartyfh/LLM-Uncertainty-Bench/tree/main`. We do not modify the underlying model predictions.

**Training details.** For each model `model`, we load logits from `outputs_base/{model}_mmlu_10k_base_icl1.pkl` using a single prompt method (`base`) and one in-context configuration (`icl1`). We exclude a small set

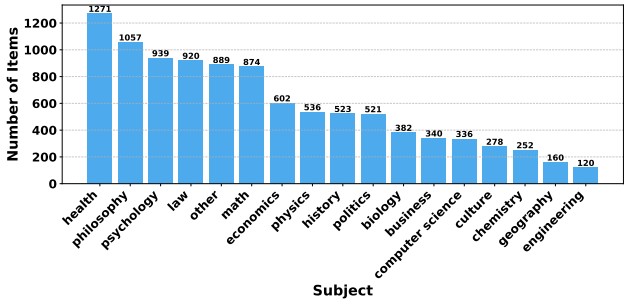

*Figure 4.* Distribution of MMLU instances across 17 subjects.

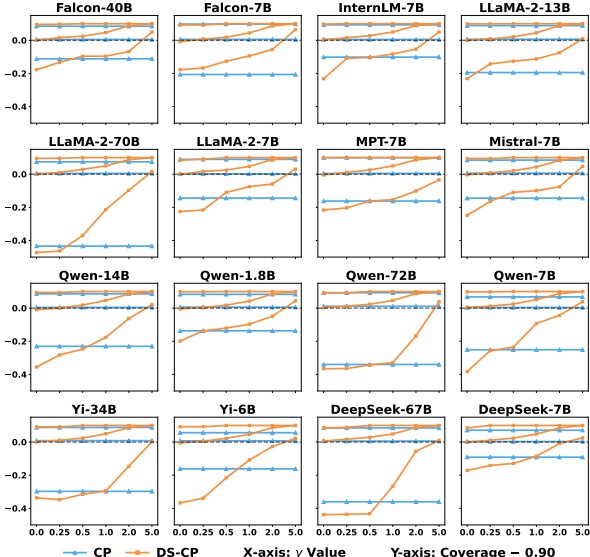

*Figure 5.* Empirical coverage of CP and DS-CP with different regularization parameters across models.

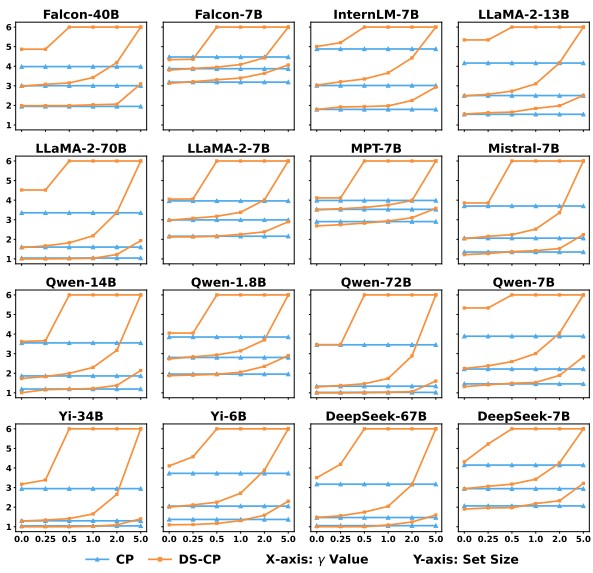

*Figure 6.* Average prediction set size of CP and DS-CP with different regularization parameters across models.

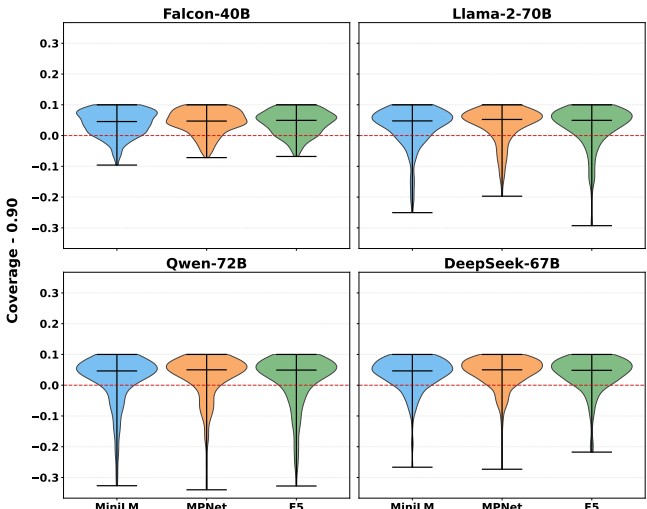

*Figure 7.* Empirical coverage of DS-CP with different embedding models. The dashed line denotes the target coverage level.

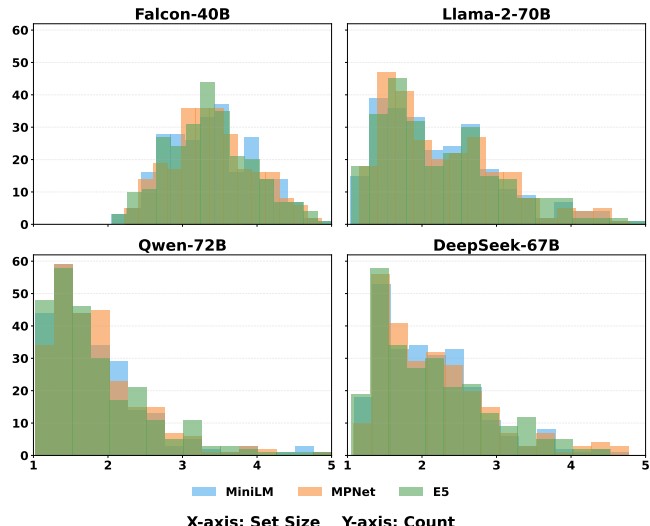

*Figure 8.* Average prediction set size of DS-CP with different embedding models.

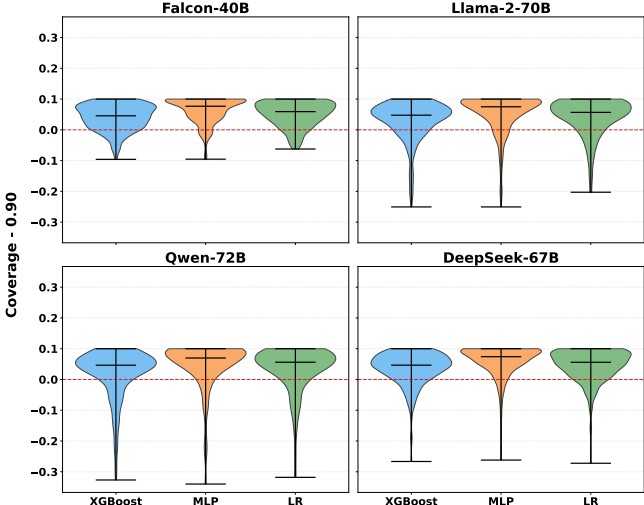

*Figure 9.* Empirical coverage of DS-CP with different classifiers. The dashed line denotes the target coverage level.

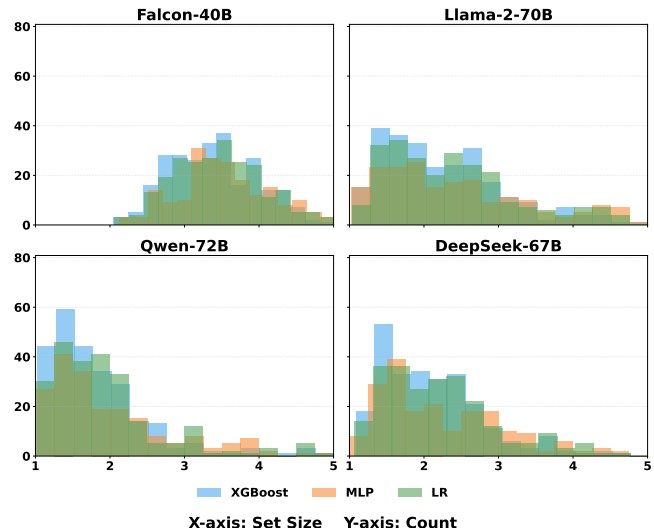

*Figure 10.* Average prediction set size of DS-CP with different classifiers.

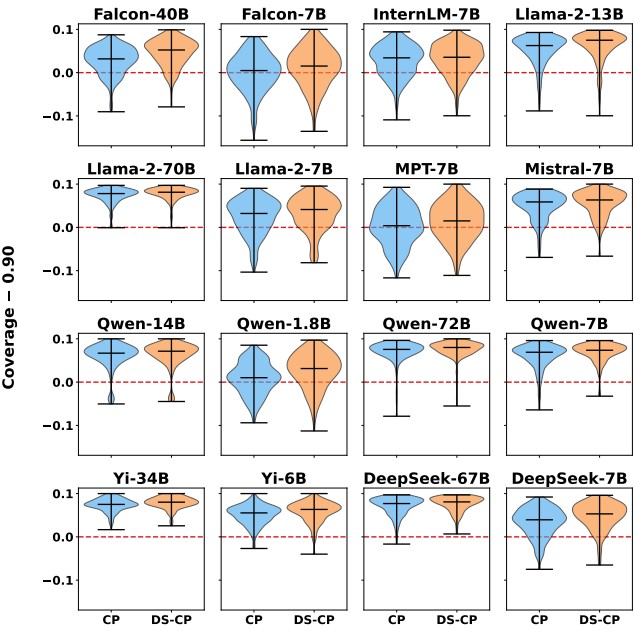

*Figure 11.* Empirical coverage of CP vs. DS-CP across models. The center bar is the median.

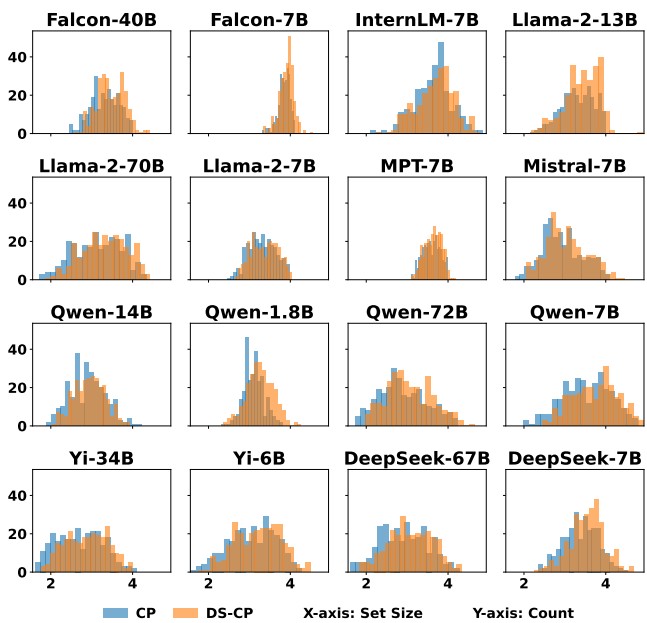

*Figure 12.* Average prediction set size of CP vs. DS-CP across models.

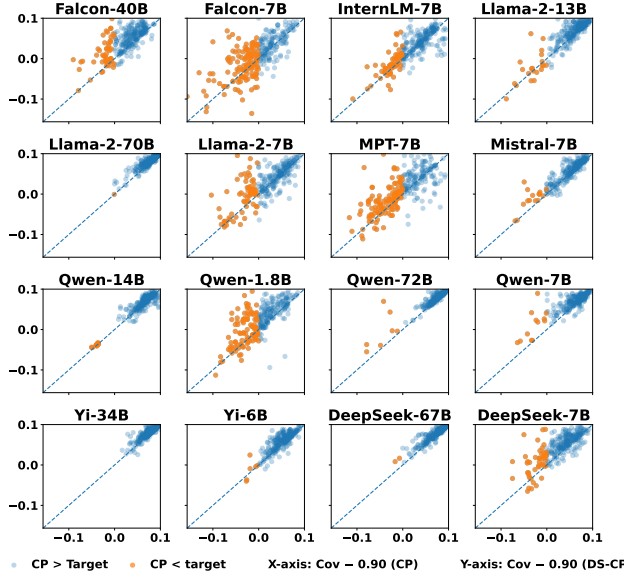

*Figure 13.* Paired coverage comparison of CP and DS-CP across models.

of item indices $[1, 3, 5, 7, 9]$ for consistency with our preprocessing. For DS-CP, we use the embedding model `all-MiniLM-L6-v2` and density ratio estimation is obtained from the XGBoost classifier trained with `max_depth`=3, `n_estimators`=50, `learning_rate`=0.2, `subsample`=0.7, `colsample_bytree`=0.7, `reg_alpha`=5, `reg_lambda`=10, `random_state`=42, `n_jobs`=1. No tuning is performed. All experiments were run locally on a MacBook Pro (14-inch, 2021) with an Apple M1 Pro chip, 16 GB RAM, and macOS Sequoia 15.5. No discrete GPU was used.

