# OpenReview forum: "Domain-Shift-Aware Conformal Prediction for Large Language Models"
_ICML.cc/2026/Conference — ICML 2026 regular_

### Official Review · Reviewer_ge5P · 2026-02-25

**Soundness:** 2
**Presentation:** 3
**Significance:** 2
**Originality:** 3
**Overall Recommendation:** 4
**Confidence:** 3

**Summary:**

The paper proposes a solution that combines density ratio estimation to deal with the reduced reliability of LLM under distribution shifts.

**Compliance With Llm Reviewing Policy:**

Affirmed.

**Final Justification:**

The authors have resolved my concerns, and I have raised my score accordingly.

**Key Questions For Authors:**

LLM is very sensitive to latency in production environments. How many milliseconds of latency does adding one step to density ratio estimation (feature extraction + classifier inference) add? How much unlabeled data from the target domain is required to train an effective weight estimator? This determines the feasibility of this method in cold start scenarios.

**Limitations:**

yes

**Strengths And Weaknesses:**

Incorporate temporal shifts (e.g., outdated knowledge) and semantic shifts (e.g., style changes) to test the generalization of density ratio estimation beyond simple task switching.

Compare against Adaptive CP  and Temperature Scaling to demonstrate the method's superiority in balancing coverage and prediction set efficiency.

Focus on Average Set Size. It is critical to prove that the method does not maintain coverage by producing excessively large, and thus impractical, prediction sets.

Analyze the sensitivity of XGBoost to various embedding models and identify failure boundaries in few-shot target domain scenarios.

Repeat experiments across different parameter scales (e.g., 7B vs. 70B) and model families (e.g., Llama, Qwen) to ensure the robustness of the findings.

---

> ### Author Rebuttal · Authors · 2026-03-30
>
> We thank the reviewer for these thoughtful comments and address them point by point below.
>
> 1. DS-CP is intended for general prompt-distribution shift rather than only subject shift. Your suggestions to study temporal shifts and semantic/style shifts are both important and well taken. These are natural and important extensions of our framework, but we believe they are beyond the scope of the present paper. We have added it to the Discussion Section.
>
> 2. We would like to clarify that the current paper already includes both the Least Ambiguous Classifier (LAC) score and the Adaptive Prediction Sets (APS) score. Therefore, our empirical study is not tied to a single conformal score construction; rather, it shows that the proposed DS-CP framework remains effective across two widely used conformal prediction scores for LLM classification tasks.
>
> Regarding temperature scaling, we view it as complementary rather than competing. Temperature scaling is a standard logit-calibration technique that rescales softmax outputs, but it does not explicitly address source-target domain mismatch. By contrast, DS-CP is designed to handle domain shift directly by estimating density ratios between the source and target prompt distributions in embedding space and incorporating these weights into conformal calibration. Since DS-CP only requires a nonconformity score as input, it can in principle be combined naturally with temperature scaling. We also emphasize an important distinction in the LLM setting: temperature is often chosen at deployment time to adjust behavior such as creativity or diversity, rather than being learned by optimizing negative log-likelihood as in standard supervised calibration pipelines. As a result, different users or deployments may choose different temperatures, which can in turn induce different prediction sets.
>
> If the reviewer is referring to Adaptive Conformal Inference Under Distribution Shift, its goal is conceptually different from ours. That line of work studies an online setting, where the distribution may evolve over time and the conformal threshold is updated sequentially to maintain long-run validity. In contrast, our paper studies an offline domain-adaptation setting for LLM prompts: we assume labeled calibration data from an old domain and unlabeled prompts from a new domain, and we explicitly estimate density ratios between the two prompt distributions after embedding them into a semantic space. Therefore, the two approaches are designed for different deployment regimes.
>
> We have added both to the Related Work Section.
>
> 3. The current paper already reports average set size (Figure 2) and a $\lambda$-ablation (Appendix B).
>
> 4. In the current version, we use a single embedding model and a fixed XGBoost configuration. In the revision, we have added a sensitivity study across embedding models and domain classifiers. Please refer to **our response to Reviewer Fzf2.**
>
> 5. The current paper already evaluates DS-CP on 16 open-source LLMs spanning 9 model families and a wide parameter range, including 1.8B to 72B models, such as Llama-2, Qwen, Falcon, Yi, DeepSeek, Mistral, MPT, Gemma, and InternLM.
>
> 6. On latency, our method is designed so that the main adaptation cost is incurred during a front-loaded calibration stage, not repeatedly during test-time prediction. In DS-CP, prompts are first embedded into a lower-dimensional semantic space, and density ratios are estimated by training a lightweight domain classifier on old- versus new-domain prompts. Importantly, both of these steps can be performed offline before calculating the prediction set threshold for a batch of target-domain prompts. After this calibration stage is completed, inference for a new test prompt only requires computing the new score; the test-point density ratio is not needed in our regularized formulation. This is exactly one of the practical advantages of our approach. In our implementation, the embedding model is a compact SentenceTransformer and the density-ratio estimator is a shallow XGBoost model with modest depth and number of trees, run on CPU with no GPU or hyperparameter tuning. This overhead is intentionally much lighter than an additional LLM forward pass.
>
> On the second question, conceptually, the amount of unlabeled target data needed is tied to how well one can separate the old and new prompt distributions in embedding space. When the shift is moderate and the embedding captures the semantic structure well, only a modest amount of unlabeled target data should be sufficient to estimate the direction of the shift. When the shift is severe, or when the embedding and density-ratio model are misspecified, more unlabeled data may be needed. This dependence is also reflected in our discussion of limitations: DS-CP relies on the quality of the embedding model and density-ratio estimator, and misspecification at that stage may reduce adaptivity or efficiency.
>
> We have clarified these practical points in the revision.

---

> > ### Author Rebuttal · Reviewer_ge5P · 2026-04-05
> >
> > The author has addressed my concerns, so I am keeping my score unchanged.

---

> > > ### Author Response · Authors · 2026-04-06
> > >
> > > Dear Reviewer ge5P,
> > >
> > > Thank you for confirming that the concerns are **Fully resolved**. We would greatly appreciate it if you could clarify what concerns currently prevent a score increase from the initial "Weak Reject", given that all previously raised issues have been addressed. If any concerns remain, please feel free to point them out, and we will respond and resolve them.
> > >
> > > Thanks again for your time.

---

### Official Review · Reviewer_NC6Y · 2026-03-09

**Soundness:** 2
**Presentation:** 2
**Significance:** 2
**Originality:** 3
**Overall Recommendation:** 3
**Confidence:** 4

**Summary:**

This paper studies conformal prediction for large language models under domain shift. The main idea is to embed prompts into a semantic space, estimate a density-ratio style weight for calibration prompts using a domain classifier, and then build a weighted conformal score distribution in that embedding space. Since direct use of the estimated test-point weight can become unstable, the paper replaces it with a regularization term

$\lambda \ge max_{1 \le i \le n} \widehat w_i,$

which yields the score distribution

$ \widehat \mu_n=\sum_{i=1}^n\frac{\widehat w_i}{\sum_{j=1}^n \widehat w_j + \lambda}\,\delta_{S_i}+\frac{\lambda}{\sum_{j=1}^n \widehat w_j + \lambda}\,\delta_{\infty}.$

The resulting prediction set contains all $y \in \mathcal Y$ such that
$S(x,y) \le Quantile(1-\alpha;\widehat \mu_n).$

The paper provides lower and upper coverage bounds expressed through total variation terms, and reports experiments on subject-shifted MMLU-style multiple-choice evaluation across 16 open-source LLMs. Empirically, the method improves coverage relative to standard conformal prediction, with a moderate increase in average set size.

**Compliance With Llm Reviewing Policy:**

Affirmed.

**Final Justification:**

I appreciate authors' responses during the rebuttal. This paper is a theoretical contribution and authors have acknowledged several of the issues I had raised during the review. However, after the rebuttal, I am not convinced if the manuscript in its current form can be accepted as it will need careful handling and revisiting of the claims. I will lower my score to weak reject.

**Key Questions For Authors:**

1. The method is described as adapting to the test prompt, yet after replacing the test-point density ratio by $\lambda$, the score distribution $\widehat \mu_n$ does not depend on $x$. Is the intended method prompt-conditional or only target-domain-conditional? If it is only domain-conditional, please revise the framing throughout. If there is a test-point-specific variant, please present it explicitly.

2. In the proof of Theorem 4.1, the set-valued map $F$ depends on random estimated weights $\widehat w_i$, but the argument controls the event difference using $\mathrm{TV}(S^i,S)$ alone. Can you provide a corrected proof, or explain why a joint discrepancy involving both weights and scores is not needed?

3. What target-domain information is assumed available when fitting the domain classifier? Algorithm 1 takes one new prompt $x$, but Step 2 requires target-domain data labeled $W=1$. Please state whether the setting is transductive, batched, or online, and quantify performance as the number of unlabeled target prompts varies.

4. Why are the experiments only compared to standard conformal prediction? Please include baselines based on weighted conformal prediction in embedding space and nonexchangeable conformal prediction with similarity-based weights.

5. Please clarify the benchmark construction. Are these experiments on raw MMLU, or on the processed uncertainty-benchmark variant derived from MMLU? The current text is hard to reconcile with the standard benchmark description, especially the number of subjects and answer options.

**Limitations:**

No. The paper discusses some technical limitations, but the limitations and societal impact discussion is not yet adequate. The impact statement is especially too thin. The paper should discuss at least the following points: failure when the density-ratio estimator is mis-specified, dependence on access to unlabeled target-domain prompts, the possibility that larger prediction sets may still be misread as strong reliability under severe shift, and the fact that the evaluation is restricted to a transductive multiple-choice setting rather than open-ended generation.

**Strengths And Weaknesses:**

### Soundness

The paper addresses a real and timely problem. Standard split conformal prediction is fragile under distribution shift, and uncertainty quantification for LLMs is an important setting where this matters. The empirical pattern reported in the paper is plausible: under subject shift, plain conformal prediction tends to under-cover, and a more conservative weighted method can recover coverage. The ablation on

$\lambda = \gamma max_{i} \widehat w_i$

is useful, and it is good that the appendix shows the expected validity-efficiency trade-off.

That said, I have several technical concerns.

First, the paper repeatedly describes the method as adapting to the test prompt or to the proximity to the test instance, but the final procedure in Algorithm 1 is not actually test-point-specific. Once the authors replace the test-point weight by a fixed regularizer $\(\lambda\)$, the empirical measure

$\widehat \mu_n = \sum_{i=1}^n\frac{\widehat w_i}{\sum_j \widehat w_j + \lambda}\,\delta_{S_i}+\frac{\lambda}{\sum_j \widehat w_j + \lambda}\,\delta_{\infty}$

no longer depends on the particular test prompt \(x\). Thus the threshold is global for the whole target domain. The only dependence on \(x\) is through the nonconformity score \(S(x,y)\), which is already true in ordinary conformal prediction. So the current method is domain-level adaptation, not local adaptation to each prompt. This matters because a large part of the narrative is built around prompt-level semantic proximity.

Second, Algorithm 1 is underspecified about what target-domain data is available at test time. Step 2 says to train a classifier with calibration data labeled \(W=0\) and test data labeled \(W=1\), but the algorithm input only contains a single new prompt \(x\). A domain classifier cannot be trained from one test prompt. The experiments appear to assume access to a batch of unlabeled target-domain prompts for each subject pair, which is a transductive setting. That is a legitimate setup, but it is materially different from online test-time prediction and should be stated much more clearly.

Third, the central validity claim feels overstated. The paper says it provides valid coverage under domain shift, but the theorems only show

$\mathbb P\left(Y_{n+1}\in \widehat C_n(X_{n+1})\right)\ge1-\alpha-\sum_{i=1}^n\frac{\widehat w_i}{\sum_{j=1}^n \widehat w_j + \lambda}\,\mathrm{TV}(S^i,S),$

plus a matching upper bound with an additional $\(\lambda\)$-term. This is not an exact finite-sample $\(1-\alpha\)$ guarantee under shift. It is an approximate bound whose slack depends on unknown total variation terms. That kind of result can still be useful, but the claims in the introduction and contribution list should be softened.

Fourth, I am not convinced by the proof as written for the data-dependent weighting claim. In the proof of Theorem 4.1, the event $\(i \in F(S)\)$ depends on the random estimated weights $\(\widehat w_1,\dots,\widehat w_n\)$, but the argument bounds the event difference using $\(\mathrm{TV}(S^i,S)\)$ rather than a joint discrepancy involving both scores and weights. Unless I missed a conditioning step, the relevant quantity seems closer to

$
\mathrm{TV}\\left((\widehat w,S^i),(\widehat w,S)\right)
$

or a conditional version given the weights. As written, the proof appears to treat $\(F\)$ as fixed while it is in fact random and data-dependent. Since the theoretical novelty of the paper is exactly this extension to data-dependent weights, this is an important issue.

Fifth, the theoretical section inherits another conceptual gap from the method design. Weighted conformal prediction under covariate shift uses the actual test-point density ratio. Here that quantity is removed and replaced by $\(\lambda\)$, which is chosen only from calibration-side estimated weights. This regularization may be practically sensible, but it also breaks the direct connection to the weighted conformal guarantee. The paper should be much clearer that it is trading exact covariate-shift correction for a stabilized heuristic with upper and lower coverage bounds.

### Presentation

The paper is readable and the high-level motivation is clear. The progression from standard conformal prediction to weighted conformal prediction and then to the proposed method is easy to follow. The appendix is also helpful in that it includes the ablation on $\(\lambda\)$, an APS variant, and implementation details.

Still, the presentation has several serious rough edges.

The largest one is the mismatch between the stated intuition and the actual algorithm. The text says the method reweights according to proximity to the test prompt, but the final threshold does not depend on the specific prompt. Relatedly, Algorithm 1 computes $\(z=h(x)\)$, yet that quantity is never used after the regularization step replaces the test-point ratio. This makes the method description feel inconsistent.

The dataset description is also confusing. The paper writes as if it is using MMLU directly, yet the numbers correspond to a processed benchmark variant rather than raw MMLU. The mention of six answer choices also conflicts with the standard benchmark description. This should be clarified, both for reproducibility and for reader trust.

The theorem statements could also be cleaned up. The notation

$\left\|\frac{\widehat w_i}{\sum_j \widehat w_j + \lambda}\right\|_{\infty}$

is misleading when the quantity is scalar. Also, Theorem 4.2 is central, but its proof is essentially omitted with the statement that it is similar to Theorem 4.1. Given that the theory is a main contribution, this is not enough.

There are some citation concerns. In particular, some references cited as density-ratio estimation methods do not actually support that claim. I encourage the authors to check this.

### Significance

The problem is important. Reliable uncertainty quantification for LLMs under shift is worth studying, and a transductive target-domain calibration setting can be practically relevant in some deployment scenarios. The empirical improvements over standard conformal prediction also suggest that the core intuition has value.

However, the significance is limited by the lack of stronger baselines and by the narrow experimental scope. The paper only compares against standard conformal prediction. For a paper whose main pitch is domain-shift-aware conformal prediction, I would have expected comparisons against at least weighted conformal prediction in the embedding space, nonexchangeable conformal prediction with similarity-based weights, and a simpler global target-domain recalibration baseline. Without these, it is hard to tell whether the reported gains come from the particular method proposed here or simply from making the procedure more conservative after observing unlabeled target prompts.

### Originality

There is some originality in combining semantic embeddings, domain classification, and conformal score weighting in the LLM uncertainty setting. The idea of stabilizing the target-side weight with a calibration-dependent \(\lambda\) is also a concrete design choice that goes beyond a direct transcription of prior weighted conformal work.

At the same time, the conceptual novelty is moderate rather than strong. Much of the method can be viewed as a domain-level reweighting variant of existing ideas from weighted or nonexchangeable conformal prediction, moved into an embedding space. That would be acceptable if the theory and evaluation were watertight, but in the current version I do not think the paper fully delivers on its stronger novelty claims.

In summary, I see a promising empirical direction and a sensible practical concern, but I also see a gap between the claims and the actual guarantees, a likely issue in the proof for data-dependent weights, and an experimental comparison that is too narrow for the paper's ambitions.

---

> ### Author Rebuttal · Authors · 2026-03-30
>
> We thank the reviewer for the careful reading, thoughtful comments, and positive assessment of the paper. We address the main concerns below and have revised the manuscript accordingly.
>
> 1. You are correct that our method is domain-level adaptation, not local adaptation to each
> prompt. Once the test-point density ratio is replaced by $\lambda$, the empirical score distribution becomes target-domain-specific but no longer depends on the individual test prompt $x$. The only remaining dependence on $x$ is through the nonconformity score $S(x,y)$, as in standard conformal prediction. We have revised the paper to emphasize that the current method is a domain-level adaptive method, rather than a prompt-level local method.
>
> 2. We agree that Algorithm 1 is ambiguous about the target-domain information available at test time. Our experiments are conducted in a batched transductive setting, not a fully online single-instance setting. For each ordered source-target subject pair, we assume access to (i) labeled calibration data from the source domain and (ii) a batch of unlabeled target-domain prompts. These unlabeled target prompts, together with the source calibration prompts, are used to train the domain classifier in embedding space and estimate the density-ratio weights; no target labels are used. This matches our implementation. We have revised the setup and Algorithm 1 to make the transductive assumption explicit, include an unlabeled target batch in the inputs, and distinguish this setting from the online case. We have also added a discussion of an online extension based on periodically updating the domain classifier with accumulated unlabeled target prompts.
>
> 3. We agree that our theoretical claims should be stated more carefully. Theorems 4.1–4.2 do not establish exact finite-sample $(1-\alpha)$ validity under arbitrary domain shift. Rather, they provide finite-sample lower and upper coverage bounds whose slack depends on weighted TV discrepancies between calibration and test score distributions, with an additional regularization term in the upper bound. This is also the nature of existing nonexchangeable CP results. We have revised the paper to state that DS-CP provides coverage bounds under domain shift, with exact validity recovered in the exchangeable case.
>
> 4. We thank the reviewer for identifying this important subtlety. The estimated weights are random, so the proof must account for this dependence. Our revision fixed this issue and added new theoretical results with discussions. Please refer to **our response to reviewer sdhg** for details.
>
> 5. We agree that the regularized procedure is not equivalent to standard weighted conformal prediction under covariate shift. Replacing the test-point ratio by $\lambda$ is a deliberate stabilization step that sacrifices the direct weighted-CP interpretation in order to avoid extreme mass on the test point and overly conservative or degenerate sets in high-dimensional embedding space. We have clarified that DS-CP is a regularized method inspired by weighted CP and nonexchangeable CP, with explicit upper/lower coverage bounds.
>
> 6. We removed the unused quantity $z=h(x)$ from the regularized algorithm. We also agree that the benchmark description should be clearer: our experiments use a processed benchmark derived from MMLU, reusing prompts/logits from the LLM-Uncertainty-Bench repository rather than raw MMLU directly. We have clarified this in the main text, including preprocessing and answer-option format.
>
> 7. We have cleaned up the notation and revised the theorem statements to reflect the updated bounds. We have also carefully audited the references and removed or replaced citations that do not directly support the relevant methodological claims, especially in the discussion of density-ratio estimation.
>
> 8. We agree that the experimental comparisons should be broader. In the revision, we have added embedding-space weighted CP using the estimated test-point ratio directly and compared it systematically with DS-CP. In our preliminary experiments, this baseline was substantially more conservative, which motivated the regularization step, and we have documented this clearly. We have also expanded the limitations and impact discussion, including dependence on unlabeled target prompts, sensitivity to density-ratio estimation, possible over-interpretation of larger prediction sets under severe shift, and the restriction to a transductive multiple-choice setting rather than open-ended generation. Please also refer to **our response to reviewer Fzf2**.

---

> > ### Author Rebuttal · Reviewer_NC6Y · 2026-03-31
> >
> > I appreciate the authors' responses. Almost all of the points that I have raised need a careful and thorough revision, which the authors have also acknowledged such.
> >
> > Given the focus area of the paper, reading other reviewer's comments also the rebuttals, I am not still quite sure if such level of surgical changes would address my concerns at the broader scale of this particular contribution for the ICML.
> >
> > Unfortunately, **I cannot recommend accepting the paper for the ICML given that it requires major revision** at this stage. I suggest the authors kindly to address the concerns raised by the reviewers, accordingly.
> >
> > Thank you again for your time responding to my questions.

---

> > > ### Author Response · Authors · 2026-04-06
> > >
> > > Dear Reviewer NC6Y,
> > >
> > > Thank you for reviewing our paper and for your insightful comments. We have carefully incorporated your feedback and added clarifying sentences to the manuscript to better highlight our contributions.

---

### Official Review · Reviewer_Fzf2 · 2026-03-12

**Soundness:** 3
**Presentation:** 3
**Significance:** 3
**Originality:** 2
**Overall Recommendation:** 4
**Confidence:** 1

**Summary:**

This paper studies uncertainty quantification for large language models (LLMs) under domain shift using conformal prediction (CP). Standard CP provides distribution-free coverage guarantees but relies on an exchangeability assumption between calibration and test data, which is often violated in real-world LLM deployments due to domain shift. The authors propose Domain-Shift-Aware Conformal Prediction (DS-CP), a framework that adapts CP to domain shifts by leveraging semantic embeddings of prompts and density-ratio weighting.

Specifically, prompts are first embedded into a semantic representation space using a pretrained embedding model. A domain classifier is then trained to distinguish calibration prompts from test prompts, which allows estimation of density ratios in the embedding space. These ratios are used to reweight calibration samples in the conformal prediction procedure. To mitigate instability from extreme density ratios, the authors introduce a regularization parameter that controls the influence of the test point in the empirical score distribution.

The paper provides theoretical coverage bounds for the proposed method under non-exchangeable data and evaluates DS-CP on the MMLU benchmark across multiple LLM families. Empirical results show that DS-CP improves empirical coverage compared with standard conformal prediction under domain shift, while maintaining relatively modest increases in prediction set size. Overall, the authors analyze a major question in uncertainty quantification for LLMs: how to preserve reliable coverage guarantees when calibration and deployment domains differ.

**Compliance With Llm Reviewing Policy:**

Affirmed.

**Final Justification:**

All of my previous concerns have been adequately addressed. Although my familiarity with conformal prediction is limited, which prevents me from providing deeper technical insights, I have a very positive overall impression of this work. I am happy to maintain my original score.

**Key Questions For Authors:**

1. The proposed method relies on semantic embeddings of prompts. How sensitive is DS-CP to the choice of embedding model (e.g., MiniLM vs. other sentence transformers)? Would different embedding spaces significantly change the density ratio estimation and resulting coverage?
2. The density ratio is estimated using a domain classifier trained to distinguish calibration and test prompts. How robust is the method to miscalibration or errors in this classifier? Have the authors evaluated alternative density-ratio estimation methods?

**Limitations:**

Yes

**Strengths And Weaknesses:**

# Strength

## Relevant and timely problem.
The paper addresses the important problem of uncertainty quantification for LLMs under domain shift. Since LLMs are frequently deployed in environments different from their calibration data, maintaining reliable predictive uncertainty is a significant challenge. The authors strive to consider the theme of robustness of conformal prediction when the exchangeability assumption is violated.

## Conceptually intuitive approach.
The proposed DS-CP framework is conceptually straightforward and builds upon existing ideas in conformal prediction under covariate shift. Embedding prompts into a semantic space and reweighting calibration points based on estimated density ratios is a natural way to adapt CP to high-dimensional textual inputs.

## Combination of theory and experiments.
The paper provides theoretical bounds on coverage under non-exchangeable data with data-dependent weights. While the bounds depend on total variation distances that may be difficult to estimate in practice, the analysis still offers useful insight into the behavior of the proposed method.

## Broad empirical evaluation across models.
The experiments include multiple LLM families and systematically test cross-domain calibration using subject pairs in MMLU. The evaluation demonstrates that standard CP frequently under-covers under domain shift and that DS-CP improves empirical coverage.

# Weaknesses

## Limited novelty relative to prior work.
The core method largely combines existing components: conformal prediction under covariate shift, density-ratio weighting, and semantic embeddings. While the integration is reasonable, the methodological novelty appears incremental. Several prior works have explored weighted or non-exchangeable conformal prediction, including approaches using similarity-based weighting for LLMs.

## Dependence on embedding quality and density ratio estimation.
The approach relies heavily on the quality of the prompt embeddings and the accuracy of the domain classifier used for density ratio estimation. The paper acknowledges this issue but does not provide an empirical study evaluating sensitivity to embedding models or classifier choices.

---

> ### Author Rebuttal · Authors · 2026-03-30
>
> We thank the reviewer for raising these important points on robustness check and for the positive assessment of our paper.
>
> **On novelty relative to prior work.** We agree that our method is not intended to introduce an entirely new conformal prediction primitive. Rather, our contribution is a principled integration of three ingredients that, to the best of our knowledge, have not previously been combined in this form for LLM domain shift:
> (i) semantic embeddings of prompts to make shift estimation feasible in high-dimensional text settings,
> (ii) density-ratio-based weighting in that semantic space, and
> (iii) a regularized conformal construction with data-dependent weights that admits finite-sample coverage guarantees and recovers standard conformal prediction in the exchangeable case. A key distinction from prior similarity-weighted or heuristic nonexchangeable conformal methods for LLMs is that our weights are not selected ad hoc from a similarity kernel. Instead, they are derived systematically from estimated density ratios and coupled with the regularization condition $\lambda \ge \max_i \hat w_i$, which is directly tied to our theoretical analysis. On the theoretical side, we develop new theoretical results with discussions tailored to the LLM setting; this extension is nontrivial relative to prior results, as discussed further in **our response to Reviewer sdhg**. Empirically, we also show across multiple experiments that this principled framework performs well in practice.
>
> **On sensitivity to embeddings and density-ratio estimation.** Your comment was also acknowledged in the Discussion section. In the current version, we use all-MiniLM-L6-v2 embeddings and an XGBoost domain classifier without extensive tuning. We have also conducted preliminary experiments with multiple embedding models and domain classifiers, and observed qualitatively similar behavior. In the revision, we have added the results from the systematic robustness study across embedding models and density-ratio estimators, and reported how these choices affect the estimated weights, empirical coverage, and average set size. Specifically, we evaluated DS-CP with several embedding backbones, including MiniLM, MPNet, and E5, together with several domain classifiers, including logistic regression, XGBoost, and a small MLP. We reported both predictive performance metrics, such as coverage and set size, and diagnostics on the variability of the estimated calibration weights.
>
> To clarify our contribution more explicitly, we have revised the end of the contribution section to state: *"Our contribution is a principled adaptation of conformal prediction under domain shift to LLM settings, where prompt distributions are high-dimensional and unstructured. The main novelty lies in combining semantic prompt embeddings with density-ratio-based weighting and a regularized data-dependent conformal construction that admits finite-sample coverage bounds and reduces to standard conformal prediction under exchangeability."*
>
> We have also added the robustness analysis described above to the appendix, and, if space permits, summarize the main findings in the main paper as well.
>
> Furthermore, we would like to emphasize, and have clarified in the paper: *``DS-CP is reasonably robust to imperfect choices of the embedding model and domain classifier. If these components are imperfect, the estimated density ratios may be noisy or less informative, which can reduce the adaptivity advantage of DS-CP. In the extreme case where the estimated weights become nearly uniform, the induced empirical score distribution is close to that of standard CP, so DS-CP correspondingly behaves similarly to standard CP. More generally, misspecification may lead to suboptimal reweighting rather than catastrophic failure, and our empirical results support this interpretation: across many model families, DS-CP shows consistently improved coverage with only modest increases in set size, indicating that the method is fairly robust to the specific embedding/classifier choices used in practice."*

---

> > ### Author Rebuttal · Reviewer_Fzf2 · 2026-03-31
> >
> > All of my previous concerns have been adequately addressed. Although my familiarity with conformal prediction is limited, which prevents me from providing deeper technical insights, I have a very positive overall impression of this work. I am happy to maintain my original score.

---

> > > ### Author Response · Authors · 2026-04-06
> > >
> > > Dear Reviewer Fzf2,
> > >
> > > We sincerely thank you for confirming that all concerns have been fully resolved, and for your strong endorsement and positive evaluation of our work.

---

### Official Review · Reviewer_sdhg · 2026-03-13

**Soundness:** 3
**Presentation:** 3
**Significance:** 3
**Originality:** 3
**Overall Recommendation:** 5
**Confidence:** 4

**Summary:**

The authors propose Domain-Shift-Aware Conformal Prediction (DSCP), a novel framework designed to adapt conformal prediction (CP) to Large Language Models (LLMs) under domain shift. Recognizing the high-dimensional and unstructured nature of LLM prompts, the method leverages sentence embeddings to project prompts into a lower-dimensional semantic space. Within this space, calibration samples are systematically reweighted based on their proximity to test prompts. The paper provides theoretical guarantees for valid coverage beyond standard exchangeability and empirically evaluates the approach on the MMLU benchmark. The results demonstrate that DSCP achieves more reliable uncertainty quantification compared to standard CP under significant distribution shifts, while maintaining computational efficiency for practical deployment.

**Compliance With Llm Reviewing Policy:**

Affirmed.

**Key Questions For Authors:**

Regarding the proof of Theorem 4.1, the current relaxation and bounding of the inequalities appear somewhat loose and straightforward. Is it possible to derive tighter bounds or obtain more refined theoretical guarantees, perhaps under specific, mild assumptions about the data distribution or the embedding space?

**Limitations:**

See the weakness part

**Strengths And Weaknesses:**

Strengths:
Well-motivated Framework: The paper addresses a highly relevant and timely problem—quantifying uncertainty in LLMs under domain shifts. The proposed DSCP framework is logically sound and practically highly feasible.
Clear Empirical Evidence: The experimental results are clearly presented, and the accompanying visualizations are highly readable, effectively supporting the authors' claims.
Weaknesses:
Limited Theoretical Novelty: While the framework is solid, the theoretical contributions appear somewhat incremental. The theoretical proof (Theorem 4.1) does not appear to pose significant technical hurdles or introduce profound mathematical innovations, leaning more towards standard applications of existing non-exchangeable CP principles.

---

> ### Author Rebuttal · Authors · 2026-03-30
>
> We thank the reviewer for raising this important point about the theoretical results and the positive rating of our paper. To address your comment, we have made progress on sharpening the analysis by exploiting the additional structure introduced by the embedding step. We believe this refinement would make the theoretical contribution more substantive, because it goes beyond a direct application of generic non-exchangeable CP theory and instead ties the guarantee directly to the embedding-based construction that is specific to DS-CP. We include the major revisions below.
>
> Define $\pi_i = \frac{\hat w_i}{\sum_{j=1}^n \hat w_j + \lambda}$ for $i=1, \ldots, n$ and $\pi_{n+1} = \frac{\lambda}{\sum_{j=1}^n \hat w_j + \lambda}$.
>
> **New Theorem 4.1**: Suppose $\lambda \ge \max_{i=1,\ldots,n}\hat w_i$ almost surely. Then $\mathbb P \bigl(Y_{n+1}\in \hat C_n(X_{n+1})\bigr) \ge 1-\alpha-\sum_{i=1}^n \mathbb E\left[\pi_i \{\rm TV}(\mathbf S^i,\mathbf S\mid \mathcal G)\right]$, where $\mathcal G=\sigma(\hat w_1,\ldots,\hat w_n,\lambda)$.
>
> Theorem 4.1 extends the nonexchangeable conformal analysis to our setting with data-dependent weights. It shows that DS-CP achieves the nominal level $1-\alpha$ up to a coverage gap determined by two factors: the normalized weights $\pi_i$, which determine how strongly each calibration point influences the procedure, and the conditional total variation distances ${\rm TV}(\mathbf S^i,\mathbf S\mid\mathcal G)$, which measure how different the score process is from the ideal exchangeable case after swapping the $i$th calibration score with the test score.
>
> Theorem 4.1 also recovers standard CP as a special case. If $\hat w_1=\cdots=\hat w_n=1$ and $\lambda=\max_{i=1,\ldots,n} \hat w_i = 1$, then DS-CP reduces exactly to standard CP, and $\mathcal G$ is trivial. If in addition $S_1,\ldots,S_{n+1}$ are exchangeable, then for every $i$ we have ${\rm TV}(\mathbf S^i,\mathbf S\mid \mathcal G)={\rm TV}(\mathbf S^i,\mathbf S)=0$.
> Hence the coverage gap vanishes and Theorem 4.1 reduces to $P\bigl(Y_{n+1}\in \hat C_n(X_{n+1})\bigr)\ge 1-\alpha$, which is exactly the usual finite-sample guarantee of standard CP. This confirms that DS-CP does not lose validity in the exchangeable setting; rather, it interpolates between standard CP and a domain-shift-aware weighted procedure.
>
> Theorem 4.1 is stated in terms of a joint total variation distance on the full score vector. The next result shows that, under a conditional independence structure, this joint discrepancy can be reduced to a simpler comparison between the conditional laws of individual scores.
>
> **Theorem 4.2**: Assume that, almost surely, for each $i=1,\ldots,n+1$, $S_i \perp (Z_1,\ldots,Z_{i-1},Z_{i+1},\ldots,Z_{n+1}) \mid Z_i$, and that $S_1,\ldots,S_{n+1}$ are conditionally independent given $Z_1,\ldots,Z_{n+1}$. Then $\sum_{i=1}^n \mathbb E\left[\pi_i {\rm TV}(\mathbf S^i,\mathbf S\mid \mathcal G)\right] \le 2\sum_{i=1}^n \mathbb E\left[\pi_i {\rm TV}(S_i\mid Z_i, S_{n+1}\mid Z_{n+1})\right]$.
>
> Theorem 4.2 clarifies what aspect of domain shift matters for coverage. The coverage gap can be controlled by how different the one-dimensional conditional score distributions are at $Z_i$ and $Z_{n+1}$. This gives a clean interpretation: DS-CP is reliable whenever nearby or highly weighted prompts induce similar score distributions, even if the original prompts themselves come from substantially different domains.
>
> The next theorem makes this idea quantitative through a smoothness condition in the embedding space.
>
> **Theorem 4.3**: Assume there exists a probability kernel $\{Q_z:z\in\mathcal Z\}$ such that $S_i\mid Z_i=z \sim Q_z \text{almost surely for all } i=1,\ldots,n+1$, and that for some metric $d(\cdot,\cdot)$ on $\mathcal Z$ and some constant $L>0$, ${\rm TV}(Q_z,Q_{z'}) \le L d(z,z') \text{for all } z,z'\in\mathcal Z$. Then $\sum_{i=1}^n \mathbb E\left[\pi_i {\rm TV}(S_i\mid Z_i, S_{n+1}\mid Z_{n+1})\right] \le L\sum_{i=1}^n \mathbb E\left[\pi_i d(Z_i,Z_{n+1})\right]$.
>
> Combining Theorems 4.1-4.3, we obtain the following qualitative picture. The coverage gap is small when the weighted calibration embeddings are close to the test embedding and, more fundamentally, when the score law varies smoothly across the embedding space. In other words, DS-CP does not require the raw prompt distributions to be close. What it requires is that the nonconformity score be stable with respect to the semantic representation.
>
> This observation is particularly relevant for LLM applications. In high-dimensional text domains, prompts from different subjects or tasks may look very different at the input level, so classical covariate-shift arguments based on raw features can be pessimistic. However, if the embedding map places semantically similar prompts nearby, and if the score function behaves similarly on nearby embeddings, then the effective coverage gap can still be small. This is exactly the setting DS-CP is designed to exploit.

---

> > ### Author Rebuttal · Reviewer_sdhg · 2026-04-04
> >
> > Thank you for your reply. I think it is indeed a good job after reading your rebuttal. I still recommend AC accept this work and wish you polish your camera ready vision.

---

> > > ### Author Response · Authors · 2026-04-06
> > >
> > > Dear Reviewer sdhg,
> > >
> > > We sincerely thank you for confirming that all concerns have been fully resolved, and for your strong endorsement and positive evaluation of our work.

---

### Decision · Program_Chairs · 2026-04-30

**Decision:**

Accept (regular)

**Comment:**

The paper aims to address the timely and important problem of maintaining reliable uncertainty quantification in LLMs when the exchangeability assumption is violated due to domain shifts. The proposed DSCP framework is well-motivated and intuitive. It adapts conformal prediction under covariate shift, handling high-dimensional textual inputs using semantic embeddings. The advantage over standard methods is supported by empirical evidence. A few issues in the presentation and experiments were pointed out by the reviewers, most of which have been addressed by the author(s). They should be incorporated in the revision. Overall, the paper makes valuable contributions and should clearly be accepted.